# LIPSCHITZ-BOUNDED EQUILIBRIUM NETWORKS

## ABSTRACT

This paper introduces new parameterizations of equilibrium neural networks, i.e. networks defined by implicit equations. This model class includes standard multilayer and residual networks as special cases. The new parameterization admits a Lipschitz bound during training via unconstrained optimization: no projections or barrier functions are required. Lipschitz bounds are a common proxy for robustness and appear in many generalization bounds. Furthermore, compared to previous works we show well-posedness (existence of solutions) under less restrictive conditions on the network weights and more natural assumptions on the activation functions: that they are monotone and slope restricted. These results are proved by establishing novel connections with convex optimization, operator splitting on non-Euclidean spaces, and contracting neural ODEs. In image classification experiments we show that the Lipschitz bounds are very accurate and improve robustness to adversarial attacks.

## 1 INTRODUCTION

Deep neural network models have revolutionized the field of machine learning: their accuracy on practical tasks such as image classification and their scalability have led to an enormous volume of research on different model structures and their properties (LeCun et al., 2015). In particular, deep residual networks with skip connections He et al. (2016) have had a major impact, and neural ODEs have been proposed as an analog with "implicit depth" (Chen et al., 2018). Recently, a new structure has gained interest: *equilibrium networks* (Bai et al., 2019; Winston & Kolter, 2020), a.k.a. *implicit deep learning models* (El Ghaoui et al., 2019), in which model outputs are defined by implicit equations incorporating neural networks. This model class is very flexible: it is easy to show that includes many previous structures as special cases, including standard multi-layer networks, residual networks, and (in a certain sense) neural ODEs.

However model flexibility in machine learning is always in tension with model *regularity* or *robustness*. While deep learning models have exhibited impressive generalisation performance in many contexts it has also been observed that they can be very brittle, especially when targeted with adversarial attacks (Szegedy et al., 2014). In response to this, there has been a major research effort to understand and certify robustness properties of deep neural networks, e.g. Raghunathan et al. (2018a); Tjeng et al. (2018); Liu et al. (2019); Cohen et al. (2019) and many others. Global Lipschitz bounds (a.k.a. incremental gain bounds) provide a somewhat crude but nevertheless highly useful proxy for robustness (Tsuzuku et al., 2018; Fazlyab et al., 2019), and appear in several analyses of generalization (e.g. (Bartlett et al., 2017; Zhou & Schoellig, 2019)).

Inspired by both of these lines of research, in this paper we propose new parameterizations of equilibrium networks with guaranteed Lipschitz bounds. We build directly on the monotone operator framework of Winston & Kolter (2020) and the work of Fazlyab et al. (2019) on Lipschitz bounds.

The main contribution of our paper is the ability to enforce tight bounds on the Lipschitz constant of an equilibrium network during training with essentially *no extra computational effort*. In addition, we prove existence of solutions with less restrictive conditions on the weight matrix and more natural assumptions on the activation functions via novel connections to convex optimization and contracting dynamical systems. Finally, we show via small-scale image classification experiments that the proposed parameterizations can provide significant improvement in robustness to adversarial attacks with little degradation in nominal accuracy. Furthermore, we observe small gaps between certified Lipschitz upper bounds and observed lower bounds computed via adversarial attack.

## 2 RELATED WORK

**Equilibrium networks, Implicit Deep Models, and Well-Posedness.** As mentioned above, it has been recently shown that many existing network architectures can be incorporated into a flexible model set called an equilibrium network (Bai et al., 2019; Winston & Kolter, 2020) or implicit deep model (El Ghaoui et al., 2019). In this unified model set, the network predictions are made not by forward computation of sequential hidden layers, but by finding a solution to an implicit equation involving a single layer of all hidden units. One major question for this type of networks is its well-posedness, i.e. the existence and uniqueness of a solution to the implicit equation for all possible inputs. El Ghaoui et al. (2019) proposed a computationally verifiable but conservative condition on the spectral norm of hidden unit weight. In Winston & Kolter (2020), a less conservative condition was developed based on monotone operator theory. Similar monotonicity constraints were previously used to ensure well-posedness of a different class of implicit models in the context of nonlinear system identification (Tobenkin et al., 2017, Theorem 1). On the question of well-posedness, our contribution is a more flexible model set and more natural assumptions on the activation functions: that they are monotone and slope-restricted.

**Neural Network Robustness and Lipschitz Bounds.** The Lipschitz constant of a function measures the worst-case sensitivity of the function, i.e. the maximum "amplification" of difference in inputs to differences in outputs. The key features of a good Lipschitz bounded learning approach include a tight estimation for Lipschitz constant and a computationally tractable training method with bounds enforced. For deep networks, Tsuzuku et al. (2018) proposed a computationally efficient but conservative approach since its Lipschitz constant estimation method is based on composition of estimates for different layers, while Anil et al. (2019) proposed a combination of a novel activation function and weight constraints. For equilibrium networks, El Ghaoui et al. (2019) proposed an estimation of Lipschitz bounds via input-to-state (ISS) stability analysis. Fazlyab et al. (2019) estimates for deep networks based on incremental quadratic constraints and semidefinite programming (SDP) were shown to give state-of-the-art results, however this was limited to analysis of an already-trained network. The SDP test was incorporated into training via the alternating direction method of multipliers (ADMM) in Pauli et al. (2020), however due to the complexity of the SDP the training times recorded were almost 50 times longer than for unconstrained networks. Our approach uses a similar condition to Fazlyab et al. (2019) applied to equilibrium networks, however we introduce a novel direct parameterization method that enables learning robust models via unconstrained optimization, removing the need for computationally-expensive projections or barrier terms.

## 3 PROBLEM FORMULATION AND PRELIMINARIES

### 3.1 PROBLEM STATEMENT

We consider the weight-tied network in which $x \in \mathbb{R}^d$ denotes the input, and $z \in \mathbb{R}^n$ denotes the hidden units, $y \in \mathbb{R}^p$ denotes the output, given by the following implicit equation

$$z = \sigma(Wz + Ux + b_z), \quad y = W_o z + b_y \tag{1}$$

where $W \in \mathbb{R}^{n \times n}$, $U \in \mathbb{R}^{n \times d}$, and $W_o \in \mathbb{R}^{p \times n}$ are the hidden unit, input, and output weights, respectively, $b_z \in \mathbb{R}^n$ and $b_y \in \mathbb{R}^p$ are bias terms. The implicit framework includes most current neural network architectures (e.g. deep and residual networks) as special cases. To streamline the presentation we assume that $\sigma : \mathbb{R} \to \mathbb{R}$ is a single nonlinearity applied elementwise, although our results also apply in the case that each channel has a different activation function, nonlinear or linear.

Equation (1) is called an equilibrium network since its solutions are equilibrium points of the difference equation $z^{k+1} = \sigma(Wz^k + Ux + b_z)$ or the ODE $\dot{z}(t) = -z(t) + \sigma(Wz(t) + Ux + b_z)$. Our goal is to learn equilibrium networks (1) possessing the following two properties:

- **Well-posedness:** For every input $x$ and bias $b_z$, equation 1 admits a unique solution $z$.

- **$\gamma$-Lipschitz:** It has a finite Lipschitz bound of $\gamma$, i.e., for any input-output pairs $(x_1, y_1)$, $(x_2, y_2)$ we have $\|y_1 - y_2\|_2 \le \gamma \|x_1 - x_2\|_2$.

## 3.2 PRELIMINARIES

**Monotone operator theory.** The theory of monotone operators on Euclidean space (see the survey Ryu & Boyd (2016)) has been extensively applied in the development of equilibrium networks (Winston & Kolter, 2020). In this paper, we will use the monotone operator theory on non-Euclidean spaces (Bauschke et al., 2011), in particular, we are interested in a finite-dimensional Hilbert space $\mathcal{H}$, which we identify with $\mathbb{R}^n$ equipped with a weighted inner product $\langle x, y \rangle_Q := y^\top Q x$ where $Q \succ 0$. The main benefit is that we can construct a more expressive equilibrium network set. A brief summary or relevant theory can be found in Appendix C.1; here we give some definitions that are frequently used throughout the paper. An operator is a set-valued or single-valued function defined by a subset of the space $A \subseteq \mathcal{H} \times \mathcal{H}$. A function $f : \mathcal{H} \to \mathbb{R} \cup \{\infty\}$ is proper if $f(x) < \infty$ for at least one $x$. The subdifferential and proximal operators of a proper function $f$ are defined as

$$\partial f(x) := \{g \in \mathcal{H} \mid f(y) \geq f(x) + \langle y - x, g \rangle_Q, \ \forall y \in \mathcal{H}\},$$

$$\mathbf{prox}_f^\alpha(x) := \{z \in \mathcal{H} \mid z = \arg\min_u \frac{1}{2}\|u - x\|_Q^2 + \alpha f(u)\}$$

respectively, where $\|x\|_Q := \sqrt{\langle x, x \rangle_Q}$ is the induced norm. For $n = 1$, we only consider the case of $Q = 1$. An operator $A$ is monotone if $\langle u - v, x - y \rangle_Q \geq 0$ and strongly monotone with parameter $m$ if $\langle u - v, x - y \rangle_Q \geq m\|x - y\|_Q^2$ for all $(x, u), (y, v) \in A$. The operator splitting problem is that of finding a zero of a sum of two operators $A$ and $B$, i.e. find an $x$ such that $0 \in (A + B)(x)$.

**Dynamical systems theory.** In this paper, we will also treat the solutions of (1) as equilibrium points of certain dynamical systems $\dot{z}(t) = f(z(t))$. Then, the well-posedness and robustness properties of (1) can be guaranteed by corresponding properties of the dynamical system's solution set. A central focus in robust and nonlinear control theory for more than 50 years – and largely unified by the modern theory of integral quadratic constraints (Megretski & Rantzer, 1997) – has been on systems which are interconnections of linear mappings and "simple" nonlinearities, i.e. those easily bounded in some sense by quadratic functions. Fortuitously, this characteristic is shared with deep, recurrent, and equilibrium neural networks, a connection that we use heavily in this paper and has previously been exploited by Fazlyab et al. (2019); El Ghaoui et al. (2019); Revay et al. (2020) and others. A particular property we are interested in is called *contraction* (Lohmiller & Slotine, 1998), i.e., any pair of solutions $z_1(t)$ and $z_2(t)$ exponentially converge to each other:

$$\|z_1(t) - z_2(t)\| \leq \alpha\|z_1(0) - z_2(0)\|e^{-\beta t}$$

for all $t > 0$ and some $\alpha, \beta > 0$. Contraction can be established by finding a Riemannian metric with respect to which nearby trajectories converge, which is a differential analog of a Lyapunov function. A nice property of a contracting dynamical system is that if it is time-invariant, a unique equilibrium exists and it possesses a certain level of robustness. Moreover, contraction can also be linked to monotone operators, i.e. a system is contracting w.r.t. to a constant (state-independent) metric $Q$ if and only if the operator $-f$ is strongly monotone w.r.t. $Q$-weighted inner product. We collect some directly relevant results from systems theory in Appendix C.2.

## 4 MAIN RESULTS

This section contains the main theoretical results of the paper: conditions implying well-posedness and Lipschitz-boundedness of equilibrium networks, and direct (unconstrained) parameterizations such that these conditions are automatically satisfied.

**Assumption 1.** *The activation function $\sigma$ is monotone and slope-restricted in $[0, 1]$, i.e.,*

$$0 \leq \frac{\sigma(x) - \sigma(y)}{x - y} \leq 1, \ \forall x, y \in \mathbb{R}, \ x \neq y. \tag{2}$$

**Remark 1.** *We will show below (Proposition 1 in Section 4.2) that Assumption 1 is equivalent to the assumption on $\sigma$ in Winston & Kolter (2020), i.e. that $\sigma(\cdot) = \mathbf{prox}_f^1(\cdot)$ for some proper convex function $f$. However, the above assumption is arguably more natural, since it is easily verified for standard activation functions. Note also that if different channels have different activation functions, then we simply require that they all satisfy (2).*

The following conditions are central to our results on well-posedness and Lipschitz bounds:

**Condition 1.** *There exists a $\Lambda \in \mathbb{D}^+$, with $\mathbb{D}^+$ denoting diagonal positive-definite matrices, such that $W$ satisfies*

$$2\Lambda - \Lambda W - W^T \Lambda \succ 0. \tag{3}$$

**Condition 2.** *Given a prescribed Lipschitz bound $\gamma > 0$, there exists $\Lambda \in \mathbb{D}^+$ such that $W, W_o, U$ satisfy*

$$2\Lambda - \Lambda W - W^T \Lambda - \frac{1}{\gamma} W_o^T W_o - \frac{1}{\gamma} \Lambda U U^T \Lambda \succ 0. \tag{4}$$

**Remark 2.** *Note that Condition 2 implies Condition 1 since $1/\gamma(W_o^T W_o + \Lambda U U^T \Lambda) \succeq 0$. As a partial converse, if Condition 1 holds, then for any $W_o, U$ there exist a sufficiently large $\gamma$ such that Condition 2 is satisfied.*

The main theoretical results of this paper are the following:

**Theorem 1.** *If Assumption 1 and Condition 1 hold, then the equilibrium network (1) is well-posed, i.e. for all $x$ and $b_z$, equation (1) admits a unique solution $z$. Moreover, it has a finite Lipschitz bound from $x$ to $y$.*

**Theorem 2.** *If Assumption 1 and Condition 2 hold, then the equilibrium network (1) is well-posed and has a Lipschitz bound of $\gamma$.*

As a consequence, we call (1) a *Lipschitz bounded equilibrium network* (LBEN) if its weights satisfy either (3) or (4). The full proofs appear in Appendices E.1 and E.2, but here we sketch some of the main ideas. We can represent (1) as an algebraic interconnection between linear and nonlinear parts:

$$v = Wz + Ux + b_z, \quad z = \sigma(v), \quad y = W_o z + b_y. \tag{5}$$

It can be shown that for any pair of solutions to the nonlinear part $z_a = \sigma(v_a), z_b = \sigma(v_b)$, if we define $\Delta_v = v_a - v_b$ and $\Delta_z = z_a - z_b$ then Assumption 1 implies the following:

$$\langle \Delta_v - \Delta_z, \Delta_z \rangle_\Lambda \geq 0. \tag{6}$$

for any $\Lambda \in \mathbb{D}^+$. This and Condition 1 can be used to prove global stability of a unique equilibrium of the differential equation $\dot{v} = -v + W\sigma(v) + Ux + b_z$, which proves there is a unique solution to (1) for any inputs. Next, straightforward manipulations of Condition 2 show that any pairs of inputs $x_a, x_b$ and outputs $y_a, y_b$ satisfy the following, where $\Delta_x = x_a - x_b$ and $\Delta_y = y_a - y_b$:

$$\gamma \|\Delta_x\|_2^2 - \frac{1}{\gamma} \|\Delta_y\|_2^2 \geq 2\langle \Delta_v - \Delta_z, \Delta_z \rangle_\Lambda \geq 0,$$

where the inequality comes (6). This implies the Lipschitz bound $\|\Delta_y\|_2 \leq \gamma \|\Delta_x\|_2$ .

**Remark 3.** *In Fazlyab et al. (2019) it was claimed that (6) holds with a richer (more powerful) class of multipliers $\Lambda$ which were previously introduced for robust stability analysis of systems with repeated nonlinearities (Chu & Glover, 1999; D'Amato et al., 2001; Kulkarni & Safonov, 2002). However this is not true: a counterexample was given in Pauli et al. (2020), and here we provide a brief explanation: even if the nonlinearities $\sigma(v_i)$ are repeated when considered as functions of $v_i$, their increments $\Delta_{zi} = \sigma(v_i + \Delta_{vi}) - \sigma(v_i)$ are not repeated when considered as functions of $\Delta_{vi}$, since they depend on the particular $v_i$ which generally differs between units.*

**Example 1.** *We illustrate the extra flexibility of Condition 1 compared to the condition of Winston & Kolter (2020) by a toy example. Consider $W \in \mathbb{R}^{2 \times 2}$ and take a slice near $W = 0$ of the form*

$$W = \begin{bmatrix} 0 & W_{12} \\ 0 & W_{22} \end{bmatrix}, \text{ for which we have: } \quad 2I - W - W^T = \begin{bmatrix} 2 & -W_{12} \\ -W_{12} & 2 - 2W_{22} \end{bmatrix}. \tag{7}$$

*By Sylvester's criterion, this matrix is positive-definite if and only if $W_{22} < 1$ and $\det(2I - W - W^T) = 4(1 - W_{22}) - W_{12}^2 > 0$, which defines a parabolic region in the $W_{12}, W_{22}$ plane.*

*Applying our condition (3), without loss of generality take $\Lambda = \text{diag}(1, \alpha)$ with $\alpha > 0$ and we have*

$$2\Lambda - \Lambda W - W^T \Lambda = \begin{bmatrix} 2 & -W_{12} \\ -W_{12} & 2\alpha - 2\alpha W_{22} \end{bmatrix}.$$

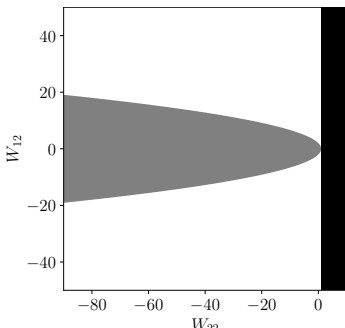

Figure 1: Valid coefficient ranges for Example 1.

*Gray region:* the condition from Winston & Kolter (2020) is feasible: $2I - W - W^T \succ 0$.

*White region (including gray region):* our well-posedness condition is feasible: $\exists \Lambda \in \mathbb{D}^+ : 2\Lambda - \Lambda W - W^T \Lambda \succ 0$.

*Black region:* neither condition feasible.

*The positivity test is now $W_{22} < 1$ and $4\alpha(1 - W_{22}) - W_{12}^2 > 0$. For each $W_{12}$ there is sufficiently large $\alpha$ such that the second condition is satisfied, since the first implies $1 - W_{22} > 0$. Hence the only constraint on $W$ is that $W_{22} < 1$, which yields a much larger region in the $W_{12}, W_{22}$ plane (see Figure 1). Interestingly, in this simple example with ReLU activation, the condition $W_{22} < 1$ is also a necessary condition for well-posedness (El Ghaoui et al., 2019, Theorem 2.8).*

### 4.1 DIRECT PARAMETERIZATION FOR UNCONSTRAINED OPTIMIZATION

Training a network that satisfies Condition 1 or 2 can be formulated as an optimization problem with convex constraints. In fact, Condition 1 is a linear matrix inequality (LMI) in the variables $\Lambda$ and $\Lambda W$, from which $W$ can be determined uniquely. Similarly, via Schur complement, Condition 2 is an LMI in the variables $\Lambda, \Lambda W, \Lambda U, W_o$, and $\gamma$, from which all network weights can be determined. In a certain theoretical sense LMI constraints are tractable – Nesterov & Nemirovskii (1994) proved they are polynomial-time solvable – however for even for moderate-scale networks (e.g. $\leq 100$ activations) the associated barrier terms or projections become a major computational bottleneck.

In this paper, we propose direct parameterization that allows learning via unconstrained optimization problem, i.e. all network parameters are transformations of free (unconstrained) matrix variables, in such a way that LMI constraints (3) or (4) are automatically satisfied.

For well-posedness, i.e. Condition (1), we parameterize via the following free variables: $V \in \mathbb{R}^{n \times n}$, $d \in \mathbb{R}^n$, and skew-symmetric[1] matrix $S = -S^T \in \mathbb{R}^{n \times n}$, from which the hidden unit weight is

$$W = I - \Psi(V^T V + \epsilon I + S), \tag{8}$$

where $\Psi = \mathrm{diag}\left(e^d\right)$ and $\epsilon > 0$ is some small constant to ensure strict positive-definiteness. Then it follows from straightforward manipulations that Condition 1 holds with $\Lambda = \Psi^{-1}$ if and only if $W$ can be written as (8). When $\Psi = I$, this reduces to the parameterization of Winston & Kolter (2020).

Similarly, for a specific Lipschitz bound, i.e. Condition 2, we add to the parameterization the free input and output weights $U$ and $W_o$, and arbitrary $\gamma > 0$. We can construct

$$W = I - \Psi\left(\frac{1}{2\gamma} W_o^T W_o + \frac{1}{2\gamma}\Psi^{-1} U U^T \Psi^{-1} + V^T V + \epsilon I + S\right), \tag{9}$$

for which (4) is automatically satisfied. Again, it can easily be verified that this construction is necessary and sufficient, i.e. any $W$ satisfying (4) can be constructed via (9).

### 4.2 MONOTONE OPERATOR PERSPECTIVE

In this section, we will show that finding the solution to LBEN (1) is equivalent to solving a well-posed operator splitting problem, and hence a unique solution exists. First, we need the following observation on the activation function $\sigma$.

**Proposition 1.** *Assumption 1 holds if and only if there exists a convex proper function $f : \mathbb{R} \to \mathbb{R} \cup \{\infty\}$ such that $\sigma(\cdot) = \mathbf{prox}_f^1(\cdot)$.*

---

[1]Note that $S$ can be parameterized via its upper or lower triangular components, or via $S = N - N^T$ with $N$ free, which can be more straightforward if $W$ is defined implicitly via linear operators, e.g. convolutions.

The proof of Proposition 1 with a construction of $f$ appears in Appendix E.3, along with a list of $f$ for popular $\sigma$. It is well-known in monotone operator theory (Ryu & Boyd, 2016) that for any convex closed proper function $f$, the proximal operator $\mathbf{prox}_f^1(x)$ is monotone and non-expansive (i.e. slope-restricted in $[0, 1]$). Proposition 1 is a converse result for scalar functions.

**Remark 4.** *To our knowledge Proposition 1 is novel, however for several popular activation functions the corresponding functions $f$ were computed in Li et al. (2019) (see also Table 3 in Appendix E.4). Compared with Li et al. (2019), our work gives a necessary and sufficient conditions.*

Now we connect LBEN (1) to an operator splitting problem.

**Proposition 2.** *Finding a solution of LBEN (1) is equivalent to solving the well-posed operator splitting problem $0 \in (A + B)(z)$ with the operators*

$$A(z) = (I - W)(z) - (Ux + b_z), \quad B = \partial\mathfrak{f} \tag{10}$$

*where $\mathfrak{f}(z) := \sum_{i=1}^n \lambda_i f(z_i)$ with $\lambda_i$ as the ith diagonal element of $\Lambda$.*

The proof appears in Appendix E.4 and Theorem 1 follows directly since the above operator splitting problem has a unique solution for any $x, b_z$.

**Computing an equilibrium.** There exist various of operator splitting algorithms to compute the solution of LBEN (1), e.g., ADMM (Boyd et al., 2011) and Peaceman-Rachford splitting (Kellogg, 1969). Winston & Kolter (2020) found that Peaceman-Rachford splitting converges very rapidly when properly tuned, and our experience agrees with this.

**Gradient backpropagation.** As shown in (Winston & Kolter, 2020, Section 3.5), the gradients of the loss function $\ell(\cdot)$ can be represented by

$$\frac{\partial \ell}{\partial(\cdot)} = \frac{\partial \ell}{\partial z_\star}(I - JW)^{-1}J\frac{\partial(Wz_\star + Ux + b_z)}{\partial(\cdot)} \tag{11}$$

where $z_\star$ denotes the solution of (1), $(\cdot)$ denotes some learnable parameters in the parameterization (8) or (9), and $J \in \mathrm{D}\sigma(Wz_\star + Ux + b_z)$ with $\mathrm{D}\sigma$ as the Clarke generalized Jacobian of $\sigma$. Since $\sigma$ is piecewise differentiable, then the set $\mathrm{D}\sigma(Wz_\star + Ux + b_z)$ is a singleton almost everywhere. The following proposition reveals that (11) is well-defined, see proof in Appendix E.5.

**Proposition 3.** *The matrix $I - JW$ is invertible for all $z_\star$, $x$ and $b_z$.*

## 4.3 CONNECTIONS TO CONVEX OPTIMIZATION

Since LBEN (1) is equivalent to an operator splitting problem, an interesting question is whether it can further be connected to a convex optimization problem. Here we construct an equivalent convex problem for the LBEN whose parameterization satisfies $S = 0$.

**Proposition 4.** *If the direct parameterization (either (8) or (9)) of an LBEN satisfies $S = 0$, then for all $x$ and $b_z$, the solution of (1) is the minimizer of the following strongly convex optimization problem:*

$$\min_z \left\langle \frac{1}{2}(I - W)z - Ux - b_z, z \right\rangle_\Lambda + \mathfrak{f}(z). \tag{12}$$

The proof is in Appendix E.6. Furthermore, for an important subclass of LBEN where $\sigma$ is ReLU, it has an equivalent convex quadratic programming (QP) formulation.

**Proposition 5.** *Consider an LBEN (1) with ReLU activation. For all $x$ and $b_z$, the solution of (1) is the minimizer of the following strongly convex QP problem:*

$$\min_z \quad \frac{1}{2}z^\top Hz + p^\top z \quad \text{s.t.} \quad z \geq 0, \ (I - W)z \geq Ux + b_z \tag{13}$$

*where $H = 2\Lambda - \Lambda W - W^\top \Lambda$ and $p = -\Lambda(Ux + b_z)$.*

Note that the QP (13) also works for the case where $S$ is non-zero. The proof (see Appendix E.7) is built on the "key insights" of ReLU activation from Raghunathan et al. (2018b). This allows one to compute the solution of LBEN (1) using the many free or commercial QP solvers.

### 4.4 Contracting Neural ODEs

In this section, we will prove the existence of a solution to (1) from a different perspective: by showing it is the equilibrium of a contracting dynamical system (a "neural ODE"). We first add a smooth state $v(t) \in \mathbb{R}^n$ to avoid the algebraic loop in (5). This idea has long been recognized as helpful for well-posedness questions (Zames, 1964). We define the dynamics of $v(t)$ by the following ODE:

$$\dot{v}(t) = -v(t) + Wz(t) + Ux + b_z, \quad z(t) = \sigma(v(t)). \tag{14}$$

The well-posedness of (1) is equivalent to the existence and uniqueness of an equilibrium of (14) for all $x$ and $b_z$, which is established by the following proposition.

**Proposition 6.** *If Assumption 1 and Condition 1 hold, then the neural ODE (14) is contracting w.r.t. some constant metric $P \succ 0$.*

The proof is in Appendix E.8. Moreover, the metric $P$ can be found via semidefinite programming. The above proposition also proves that the nonlinear operator $-f$ with $f(v) = -v + W\sigma(v) + Ux + b_z$, zeros of which define solutions of LBEN (1), is actually monotone w.r.t. the $P$-weighted inner product, which gives a first-order cutting-plane oracle for the zero location $v_\star$ such that $f(v_\star) = 0$. I.e. given a test point $v_t \neq v^\star$, it proves that $v_\star$ is in the half-space defined by $\langle v_\star - v_t, f(v_t) \rangle_P > 0$. This may offer alternative ways to solve LBEN (1), e.g. via Nemirovski (2004); Nesterov (2007).

### 4.5 Feedforward Networks as a Special Case

Consider a multi-layer feedforward network of the form

$$z_1 = U_0 x + b_0, \quad z_{\ell+1} = \sigma(W_\ell z_\ell + b_\ell), \ \ell = 1, \ldots, L-1, \quad y = W_L z_L + b_L, \tag{15}$$

which can be rewritten as an equilibrium network (1) as shown in Appendix A The above equilibrium network is obviously well-posed as a unique solution exists. The following proposition shows that (44) is also an LBEN.

**Proposition 7.** *The LBEN parameterization (8) contains all feedforward networks.*

In Winston & Kolter (2020), a set of well-posed equilibrium network, called monotone operator equilibrium network (MON), is introduced via the following parameterization

$$W = (1 - m)I - A^\top A + B^\top - B \tag{16}$$

where $m > 0$ is a hyper-parameter, $A, B$ are learnable matrices. The MON parameterization can be understood as a special case of LBEN with a fixing $\Psi = I$.

**Proposition 8.** *The MON parameterization (16) does not contain all feedforward networks, and if $m \geq 1$ it does not contain any feedforward networks.*

From the proof (see Appendix E.11). The set of feedforward networks in MON shrinks as the hyper-parameter $m$ increases. Most experiments in Winston & Kolter (2020) use $m = 1$, which excludes all feedforward networks.

In the feedforward case, our Lipschitz bound condition (4) is equivalent to the state-of-art bound estimation method in Fazlyab et al. (2019). The major benefit of our direct parameterization (9) is that it allows such bounds to be imposed during training without any additional computational cost. The details are given in Appendix D.

## 5 Experiments

In this section we test our approach on the MNIST and CIFAR-10 image classification problems. Our numerical experiments focus on model robustness, the trade-off between model performance and the Lipschitz constant, and the tightness of the Lipschitz bound. We compare the proposed LBEN to unconstrained equilibrium networks, monotone operator equilibrium network (MON) of Winston & Kolter (2020), and fully connected networks trained using Lipschitz margin training (LMT) (Tsuzuku et al., 2018). When studying model robustness to adversarial attacks, we use the L2 Fast Gradient Method, implemented as part of the Foolbox toolbox (Rauber et al., 2020). All

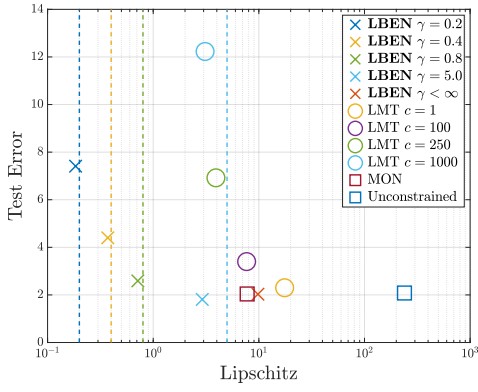 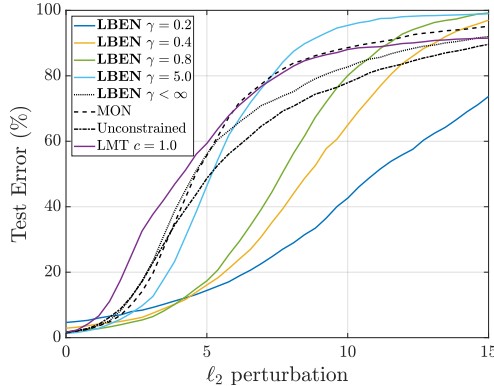

(a) Nominal test error vs Lipschitz constant estimates: markers indicate observed lower bounds for all methods, vertical lines indicate certified upper bounds for LBEN

(b) Test error with adversarial perturbation versus size of adversarial perturbation. Lower is better.

Figure 2: Image classification results on MNIST character recognition data set.

models are trained on a either a standard desktop computer with an NVIDIA GeForce RTX 2080 graphics card or using a google cloud instance with a Nvidia Tesla V100 graphics card. Details of the models and training procedure can be found in Appendix F, all code will be made available online but links are omitted due to the double-blind review process.

## 5.1 MNIST EXPERIMENTS WITH FULLY-CONNECTED NETWORKS

In Figure 2a the test error versus the observed Lipschitz constant, computed via adversarial attack for each of the models trained. We can see clearly that the parameter $\gamma$ in LBEN offers a trade-off between test error and Lipschitz constant. Comparing the $\text{LBEN}_{\gamma=5}$ with both MON and $\text{LBEN}_{\gamma<\infty}$, we also note a slight regularizing effect in the lower test error.

By comparison, LMT (Tsuzuku et al., 2018) with $c$ as a tunable regularization parameter displays a qualitatively similar trade-off, but underperforms LBEN in terms of both test error and robustness. If we examine the unconstrained equilibrium model, we observe a Lipschitz constant more than an order of magnitude higher, i.e. this model has regions of extremely high sensitivity, without gaining any accuracy in terms of test error.

For the LBEN models, the lower and upper bounds on the Lipschitz constant are very close: the markers are very close to their corresponding lines in Figure 2a, see also the table of numerical results in Appendix A in which the approximation accuracy is in many cases around 90%.

Next we tested robustness of classification accuracy to adversarial attacks of various sizes, the results are shown in Figure 2b and summarized in Table 1. We can clearly see that decreasing $\gamma$ (i.e. stronger regularization) in the LBEN models results in a far more gradual degradation of performance as perturbation size increases, with only a mild impact on nominal (zero perturbation) test error.

Next, we examined the impact of our parameterization on computational complexity compared to other equilibrium models. The test and training errors versus number of epochs are plotted in Figure 5, and we can see that all models converge similarly, and also take roughly the same amount of time per epoch. This is a clear contrast to the results of Pauli et al. (2020) in which imposing Lipschitz constraints resulted in fifty-fold increase in training time. Interestingly, we can also see in Figure 5 the effect of regularisation for LBEN with $\gamma = 5$: higher training error but lower test error. We have observed several cases where the unconstrained equilibrium model became unstable during training, LBEN never exhibits this problem.

Finally, we examined the quality of the Lipschitz bounds as a function of network size, comparing the upper and lower bounds on fully connected networks with width 20 to 1000. The results are shown in Figure 6. It can be observed that network size only has a mild effect on the quality of the Lipschitz bounds, which decrease slightly as width is increased by a factor of 50.

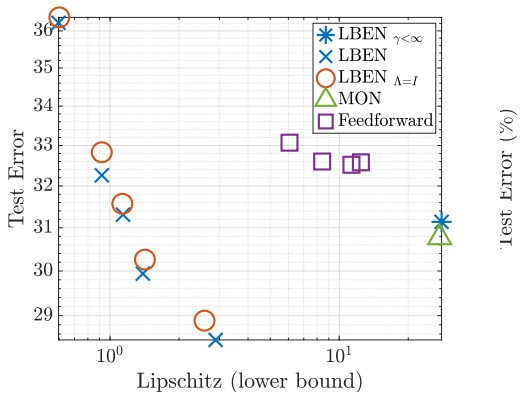 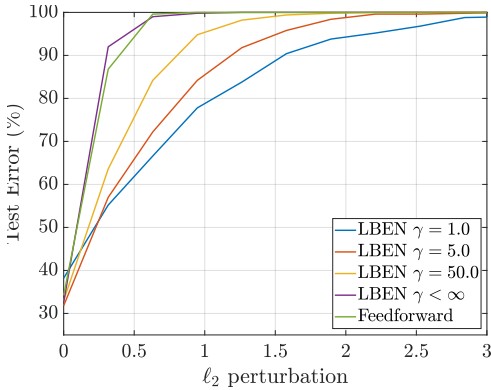

(a) Nominal test error vs observed lower bound on Lipschitz constant.

(b) Test error with adversarial perturbation versus size of adversarial perturbation. Lower is better.

Figure 3: Image classification results on CIFAR-10 data set.

## 5.2 CIFAR-10 EXPERIMENTS WITH CONVOLUTIONAL NETWORKS

The previous example looked at simple fully connected networks, however, our approach can also be applied to structured layers such as convolutions. Here, we perform several experiments exploring the use of convolutional layers on the CIFAR-10 dataset. To study the improved expressibility we will compare the LBEN to the LBEN with its metric set to the identity, denoted LBEN $_{\Lambda=I}$. Note that the model set LBEN $_{\Lambda=I,\gamma<\infty}$ corresponds to the MON. Additional model details can be found in Appendix F.2.

In Figure 3a, we have plotted the test performance versus the observed Lipschitz constant for the LBEN and LBEN $_{\Lambda=I}$ for varying Lipschitz bound $\gamma = 1, 2, 3, 5, 50$, along with the LBEN$_{\gamma<\infty}$, MON, and feed-forward convolutional networks with 40, 81, 160, and 200 channels. Again, we see that the Lipschitz bound has a regularizing effect, trading off between nominal fit and robustness. Additionally, we see that the LBEN provides both better performance and robustness than the traditional feed-forward convolutional networks of similar sizes, highlighting the benefit of the equilibrium network structure.

Comparing LBEN and LBEN$_{\Lambda=I}$, we can see that the metric gives higher quality models for LBEN with specified $\gamma$, but it is slightly worse for LBEN $\gamma < \infty$ compared to MON. This is likely due to the extra expressiveness of the model leading to some overfitting. This can also be seen in the training curves in Figure 7.

Figure 3b shows the test error versus the size of adversarial perturbation for the lBEN and 162 channel feed-forward convolutional network. We observe that the LBEN provides a much more gradual loss in performance than the feed-forward network, with $\gamma = 5$ offering an excellent mix of nominal performance and robustness. The feed-forward networks of different sizes exhibited similar results, however only one is plotted in Figure 3b for clarity.

## 6 CONCLUSIONS

In this paper, we have shown that the flexible framework of equilibrium networks can be made robust via a simple and direct parameterization which results in guaranteed Lipschitz bounds. These results can also be directly applied (as a special case) to standard multilayer and residual deep neural networks, and also provide a direct parameterization of nonlinear ODEs satisfying strong stability and robustness properties.

Extension to equilibrium network structures more general than (1) is an interesting area for future research. Our results can be extended to more general multivariable "activations" if they can be described accurately via monotonicity properties or integral quadratic constraints. One particular example where this is possible is where the "activation" computes the $\arg\min$ of a quadratic program of the sort that appears in constrained model predictive control (Heath & Wills, 2007).

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

# A    EXPERIMENTAL RESULTS ON MNIST CHARACTER RECOGNITION

This appendix contains tables of results on MNIST and CIFAR-10 data sets.

Legend:

- Err: Test error (%),
- $\|a\|_2$: $\ell^2$ norm of adversarial attack.
- $\gamma_{up}$: certified upper bound on Lipschitz constant (for models that provide one).
- $\gamma_{low}$: observed lower bound on Lipschitz constant via adversarial attack.
- $\gamma$ approx: approximation ratio of Lipschitz constant as percentage $= 100 \times \left( \frac{\gamma_{low}}{\gamma_{up}} \right)$.

Models:

- LBEN: the proposed Lipschitz bounded equilibrium network..
- MON: the monotone operator equilibrium network of Winston & Kolter (2020).
- UNC: an unconstrained equilibrium network, i.e. $W$ directly parameterized.
- LMT: Lipschitz Margin Training model as in Tsuzuku et al. (2018).
- Lip-NN: The Lipschitz Neural Network model of Pauli et al. (2020). Note these figures are as reported in (Pauli et al., 2020), all other figures are calculated by the authors of the present paper.

| Model | Err: $\|a\|_2 = 0$ | Err: $\|a\|_2 \leq 5$ | Err: $\|a\|_2 \leq 10$ | $\gamma_{up}$ | $\gamma_{low}$ | $\gamma$ approx |
|---|---|---|---|---|---|---|
| LBEN$_{\gamma<\infty}$ | 2.03 | 56.0 | 82 | - | 9.8 | - |
| LBEN$_{\gamma=5}$ | **1.81** | 46.4 | 95.4 | 5 | 2.912 | 58.2% |
| LBEN$_{\gamma=1}$ | 2.36 | 19.4 | 85.5 | 1 | 0.865 | 86.5% |
| LBEN$_{\gamma=0.8}$ | 2.59 | 17.4 | 80.1 | 0.8 | 0.715 | 89.4% |
| LBEN$_{\gamma=0.4}$ | 4.44 | 16.1 | 65.0 | 0.4 | 0.372 | 93% |
| LBEN$_{\gamma=0.2}$ | 7.41 | **14.4** | **42.6** | 0.2 | 0.184 | 92% |
| MON | 2.04 | 55.8 | 88.6 | - | 7.75 | - |
| UNC | 2.08 | 48.75 | 77.9 | - | 239.0 | - |
| LMT$_{c=1}$ | 2.3 | 59.4 | 88.1 | - | 17.5 | - |
| LMT$_{c=100}$ | 3.4 | 65.4 | 92.0 | - | 7.66 | - |
| LMT$_{c=250}$ | 6.92 | 61.8 | 98.4 | - | 6.92 | - |
| LMT$_{c=1000}$ | 12.23 | 78.4 | 98.9 | - | 3.10 | - |
| Lip-NN | 3.55 | - | - | 8.74 | - | - |

Table 1: Results from MNIST experiments.

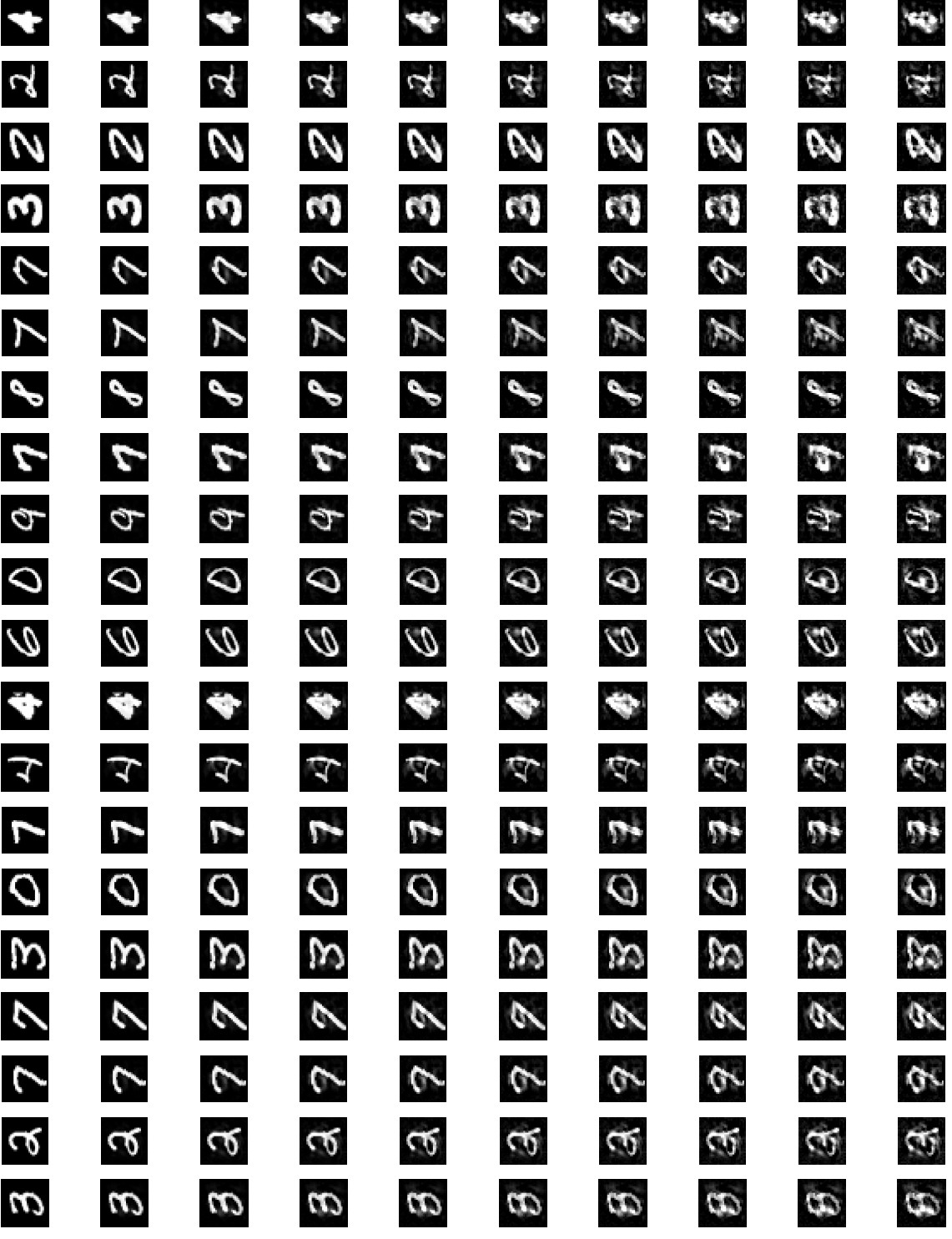

Figure 4: Random selection of MNIST adversarial examples from Figure 2b. Top to bottom is increasing perturbation size. Left to right are different examples.

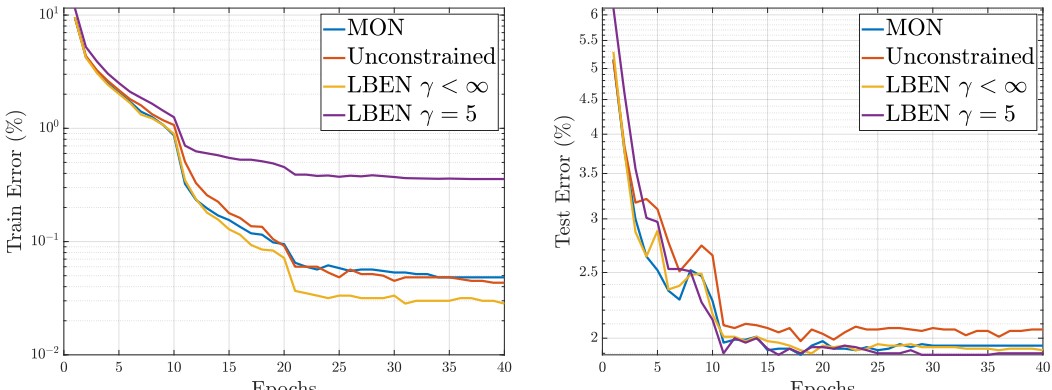

Figure 5: **Left:** Training set error versus epochs. **Right:** Test set error versus epochs. Note that the left and right plots are on different scales. The time per epoch for the MON, unconstrained, LBEN$_{\gamma<\infty}$ and LBEN$_{\gamma=5}$ networks are 14.4, 16.1, 14.9 and 14.8 seconds per epoch respectively.

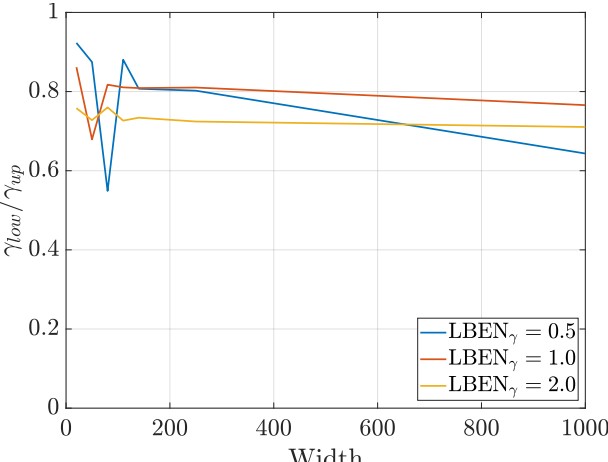

Figure 6: Approximation accuracy of the Lipschitz bound versus the network width of LBEN from the MNIST example. The certified upper bound is $\gamma_{up}$ and the observed lower bound is $\gamma_{low}$.

## B    EXPERIMENTAL RESULTS ON CIFAR-10 DATASET

| Model | Err: $\|a\|_2 = 0$ | Err: $\|a\|_2 \leq 0.5$ | Err: $\|a\|_2 \leq 1.0$ | $\gamma_{up}$ | $\gamma_{low}$ | $\gamma$ approx |
|---|---|---|---|---|---|---|
| LBEN$_{\gamma < \infty}$ | 31.1 | 96.1 | 100 | - | 31.1 | - |
| LBEN$_{\gamma=50}$ | **28.4** | 75.5 | 95.4 | 50 | 2.89 | 5.7% |
| LBEN$_{\gamma=5}$ | 29.9 | 65.8 | 85.5 | 5 | 1.39 | 27.8% |
| LBEN$_{\gamma=3}$ | 31.3 | 64.2 | 83.5 | 3 | 1.14 | 38.0% |
| LBEN$_{\gamma=2}$ | 37.9 | 62.5 | 80.5 | 2 | 0.92 | 46.0% |
| LBEN$_{\gamma=1}$ | 36.2 | **61.8** | **78.8** | 1 | 0.60 | 60.0 % |
| FF$_{W=40}$ | 33.07 | 91.5 | 99.8 | - | 6.06 | - % |
| FF$_{W=81}$ | 32.6 | 93.3 | 100 | - | 8.42 | - % |
| FF$_{W=162}$ | 32.5 | 95.0 | 100 | - | 11.3 | - % |
| FF$_{W=200}$ | 32.6 | 94.5 | 100 | - | 12.4 | - % |

Table 2: Results from CIFAR experiments. FF refers to the feed-forward convolutional network.

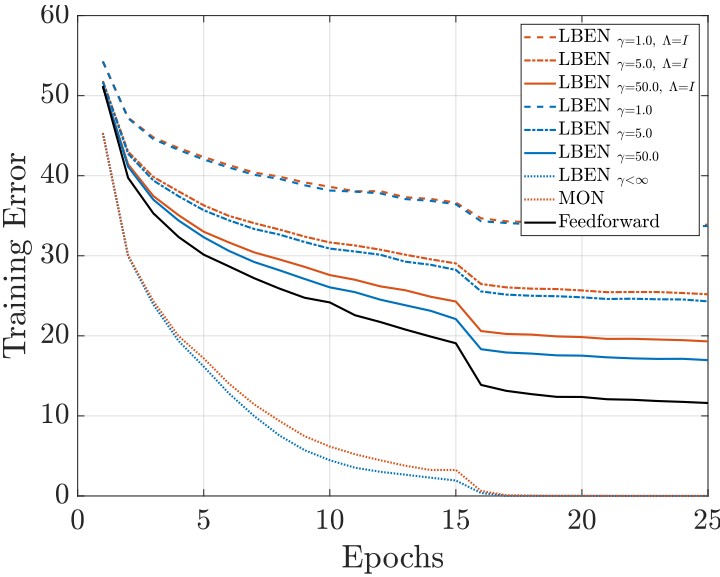

Figure 7: LBEN and MON training error versus epochs on CIFAR-10 dataset. The red curves have the metric set so that $\Lambda = I$ whereas the blue curves optimize over the metric. The line styles correspond to different gain bounds. Note that both MON and LBEN$_{\gamma<\infty}$ achieve zero training error.

## C    PRELIMINARIES

### C.1    MONOTONE OPERATORS WITH NON-EUCLIDEAN INNER PRODUCTS

We present some basic properties of monotone operators on a finite-dimensional Hilbert space $\mathcal{H}$, which we identify with $\mathbb{R}^n$ equipped with a weighted inner product $\langle x, y \rangle_Q = y^\top Q x$ with $Q \succ 0$. For $n = 1$, we only consider the case of $Q = 1$. The induced norm $\|x\|_Q$ is defined as $\sqrt{\langle x, x \rangle_Q}$. A *relation* or *operator* is a set-valued or single-valued map defined by a subset of the space $A \subseteq \mathcal{H} \times \mathcal{H}$; we use the notation $A(x) = \{y \mid (x, y) \in A\}$. If $A(x)$ is a singleton, we called $A$ a function. Some commonly used operators include: the linear operator $A(x) = \{(x, Ax) \mid x \in \mathcal{H}\}$; the operator sum $A + B = \{(x, y + z) \mid (x, y) \in A, (x, z) \in B\}$; the inverse operator $A^{-1} = \{(y, x) \mid (x, y) \in A\}$; and the subdifferential operator $\partial f = \{(x, \partial f(x))\}$ with $x = \mathbf{dom}\, f$ and $\partial f(x) = \{g \in \mathcal{H} \mid f(y) \geq f(x) + \langle y - x, g \rangle_Q, \forall y \in \mathcal{H}\}$. An operator $A$ has Lipschitz constant $L$ if for any $(x, u), (y, v) \in A$

$$\|u - v\|_Q \leq L\|x - y\|_Q. \tag{17}$$

An operator $A$ is non-expansive if $L = 1$ and contractive if $L < 1$. An operator $A$ is monotone if

$$\langle u - v, x - y \rangle_Q \geq 0, \ \forall (x, u), (y, v) \in A. \tag{18}$$

It is strongly monotone with parameter $m$ if

$$\langle u - v, x - y \rangle_Q \geq m \|x - y\|_Q^2, \ \forall (x, u), (y, v) \in A. \tag{19}$$

A monotone operator $A$ is maximal monotone if no other monotone operator strictly contains it, which is a property required for the convergence of most fixed point iterations. Specifically, an affine operator $A(x) = Wx + b$ is (maximal) monotone if and only if $QW + W^\top Q \succeq 0$ and strongly monotone if $QW + W^\top Q \succeq mI$. A subdifferential $\partial f$ is maximal monotone if and only if $f$ is a convex closed proper function.

The resolvent and Cayley operators for an operator $A$ are denoted $R_A$ and $C_A$ and respectively defined as

$$R_A = (I + \alpha A)^{-1}, \quad C_A = 2R_A - I \tag{20}$$

for any $\alpha > 0$. When $A(x) = Wx + b$, then

$$R_A(x) = (I + \alpha W)^{-1}(x - \alpha b) \tag{21}$$

and when $A = \partial f$ for some CCP function $f$, then the resolvent is given by a proximal operator

$$R_A(x) = \mathbf{prox}_f^\alpha(x) := \arg\min_z \frac{1}{2}\|x - z\|_Q^2 + \alpha f(z). \tag{22}$$

The resolvent and Cayley operators are non-expansive for any maximal monotone $A$, and are contractive for strongly monotone $A$. Operator splitting methods consider finding a zero in a sum of operators (assumed here to be maximal monotone), i.e., find $z$ such that $0 \in (A + B)(z)$. For example, the convex optimization problem in (12) can be formulated as an operator splitting problem with $A(z) = (I - W)z - b$ and $B = \partial f$. Proposition 2 shows that $A$ is strongly monotone and Lipschitz with some parameters of $m$ and $L$. Here we give some popular operator splitting methods for this problem as follows.

- Forward-backward splitting: $z^{k+1} = R_B(z^k - \alpha A(z^k))$, i.e.,

$$\begin{aligned} u^k &= ((1 - \alpha)I + \alpha W)z^k + \alpha b \\ z^{k+1} &= \mathbf{prox}_f^\alpha(u^k) \end{aligned} \tag{23}$$

- Peaceman-Rachford splitting: $u^{k+1} = C_A C_B(u^k)$, $z^k = R_B(u^k)$, i.e.,

$$\begin{aligned} u^{k+1/2} &= 2z^k - u^k, \\ z^{k+1/2} &= (I + \alpha(I - W))^{-1}(u^{k+1/2} + \alpha b), \\ u^{k+1} &= 2x^{k+1/2} - u^{k+1/2}, \\ z^{k+1} &= \mathbf{prox}_f^\alpha(u^{k+1}). \end{aligned} \tag{24}$$

- Douglas-Rachford splitting (or ADMM): $u^{k+1} = 1/2(I + C_A C_B)(u^k)$, $z^k = R_B(u^k)$, i.e.,

$$\begin{aligned} u^{k+1/2} &= 2z^k - u^k, \\ z^{k+1/2} &= (I + \alpha(I - W))^{-1}(u^{k+1/2} + \alpha b), \\ u^{k+1} &= 2x^{k+1/2} - u^{k+1/2}, \\ z^{k+1} &= \mathbf{prox}_f^\alpha(u^{k+1}). \end{aligned} \tag{25}$$

A sufficient condition for forward-backward splitting to converge is $\alpha < 2m/L^2$. The Peacemance-Rachford and Douglas-Rachford methods converge for any $\alpha > 0$, although the convergence speed will often vary substantially based upon $\alpha$.

### C.2 DYNAMICAL SYSTEM THEORY

In this section, we present some concepts and results of dynamical system theory that are used in this paper. We consider a nonlinear system of the form

$$\dot{z}(t) = f(z(t)) \tag{26}$$

where $z(t) \in \mathbb{R}^n$ is the state, and the function $f$ is assumed to be Lipschitz continuous. By Picard's existence theorem we have a unique a solution for any initial condition. The above system is time-invariant since $f$ is not explicitly depends on $t$. System (26) is called linear time-invariant (LTI) system if $f(z) = Az + b$ for some matrix $A \in \mathbb{R}^{n \times n}$ and $b \in \mathbb{R}^n$. The point $z_\star \in \mathbb{R}^n$ is call an equilibrium of (26) if $f(z_\star) = 0$.

The central concern in dynamical system theory is *stability*. While there are many different stability notions (Khalil, 2002), here we mainly focus on two of them: exponential stability and contraction w.r.t a constant metric $Q \succ 0$. System (26) is said to be locally exponentially stable at the equilibrium $z_\star$ w.r.t. to the metric $Q$ if there exist some positive constants $\alpha, \beta, \delta$ such that for any initial condition $z(0) \in \mathcal{B}_\delta(z_\star) := \{z \mid \|z - z_\star\|_Q < \delta\}$, the following condition holds:

$$\|z(t) - z_\star\| \leq \alpha \|z(0) - z_\star\|_Q e^{-\beta t}, \quad \forall t > 0. \tag{27}$$

And it is said to be globally exponentially stable if the above condition also holds for any $\delta > 0$. The exponentially stability can be verified via Lyapunov's second method, i.e., finding a Lyapunov function $V = \|z\|_P^2$ with $P \succ 0$ such that $\dot{V}(t) \leq -2\beta V(t)$ along the solutions, i.e.,

$$(z - z_\star)^\top P f(z) + f(z)^\top P(z - z_\star) + 2\beta(z - z_\star)^\top P(z - z_\star) \leq 0. \tag{28}$$

System (26) is said to be contracting w.r.t. the metric $Q$ if there exist some positive constants $\alpha, \beta$ such that for any pair of solutions $z_1(t)$ and $z_2(t)$, we have

$$\|z_1(t) - z_2(t)\|_Q \leq \alpha \|z_1(0) - z_2(0)\|_Q e^{-\beta t}, \quad \forall t > 0. \tag{29}$$

Note that contraction is a much stronger notion than global exponential stability as Condition (27) can be implied by Condition (29) by setting $z_1 = z$ and $z_2 = z_\star$. However, unlike the Lyapunov analysis, contraction analysis can be done via simple local analysis which does not require any prior-knowledge about the equilibrium $z_\star$. Specifically, contraction can be established by the local exponential stability of the associated differential system defined by

$$\dot{\Delta}_z = \mathrm{D}f(z)\Delta_z$$

where $\Delta_z(t)$ is the infinitesimal variation between $z(t)$ and its neighborhood solutions, and $\mathrm{D}f$ is Clarke generalized Jacobian. The condition for (26) to be contracting can be represented as a state-dependent Linear Matrix Inequality (LMI) as follows

$$P\mathrm{D}f(z) + \mathrm{D}f(z)^\top P + 2\beta P \prec 0 \tag{30}$$

for some $P \succ 0$ and all $z \in \mathbb{R}^n$. For an LTI system, exponential stability and contraction are equivalent and the stability condition can be s if $A$ is Hurwitz stable (i.e. all eigenvalues of $A$ have strictly negative real part).

For most applications, the dynamic system usually involves an external input $x(t) \in \mathbb{R}^m$ and an output $y(t) \in \mathbb{R}^p$, whose state-space representation takes the form of

$$\dot{z}(t) = f(z(t), x(t)), \quad y(t) = h(z(t), x(t)). \tag{31}$$

Here we measure the robustness of the above system under input perturbation by incremental $L_2$-gain. That is, system (31) has an incremental $L_2$-gain bound of $\gamma$ if for any pair of inputs $x_1(\cdot), x_2(\cdot)$ with $\int_0^T \|x_1(t) - x_2(t)\|_2^2 dt < \infty$ for all $T > 0$, and any initial conditions $z_1(0)$ and $z_2(0)$, the solutions of (31) exists and satisfy

$$\int_0^T \|y_1(t) - y_2(t)\|_2^2 \, dt \leq \gamma^2 \int_0^T \|x_1(t) - x_2(t)\|_2^2 \, dt + \kappa(z_1(0), z_{(0)}) \tag{32}$$

for some function $\kappa(z_1, z_2) \geq 0$ with $\kappa(z, z) = 0$. Note that $\gamma$ can be viewed as a Lipschitz bound of all the mappings defined by (31) with some initial condition from the input signal $x(\cdot)$ to

$y(\cdot)$. For any two constant inputs $x_1, x_2$, let $z_1, z_2$ and $y_1, y_2$ be the corresponding equilibrium and steady-state output, respectively. From (32) we have

$$\|y_1 - y_2\|_2^2 \leq \|x_1 - x_2\|_2^2 + \kappa(z_1, z_2)/T,$$

which implies a Lipschitz bound of $\gamma$ as $T \to \infty$.

A particular class of nonlinear systems that have strong connections to various neural networks is the so-called Luré system, which takes the form of

$$\dot{z}(t) = Az(t) + B\phi(Cz(t)) \tag{33}$$

where $A, B, C$ are constant matrices with proper size, and $\phi$ is a static nonlinearity with sector bounded of $[\alpha, \beta]$: for all solution $(v, w)$ with $w = \phi(v)$

$$(w - \alpha v)^\top (\beta v - w) \geq 0 \tag{34}$$

or equivalently $\begin{bmatrix} v \\ w \end{bmatrix}^\top \Pi \begin{bmatrix} v \\ w \end{bmatrix} \geq 0$ with

$$\Pi = \begin{bmatrix} 2\alpha\beta I & (\alpha + \beta)I \\ (\alpha + \beta)I & -2I \end{bmatrix}. \tag{35}$$

This implies that the origin is an equilibrium since $\phi(0) = 0$. The above system can be viewed as a feedback interconnection of a linear system

$$G : \begin{cases} \dot{z}(t) = Az(t) + Bw(t) \\ v(t) = Cz(t) \end{cases} \tag{36}$$

and a nonlinear memoryless component $w(t) = \phi(v(t))$. The above linear system can also be described by a transfer function $G(s)$ with $s \in \mathbb{C}$. We refer to Hespanha (2018) for details about frequency-domain concepts and results of linear systems. The frequency-domain representation for the sector bounded condition (34) can be written as

$$\begin{bmatrix} \hat{v}(j\omega) \\ \hat{w}(j\omega) \end{bmatrix}^* \Pi \begin{bmatrix} \hat{v}(j\omega) \\ \hat{w}(j\omega) \end{bmatrix} \geq 0 \quad \forall \omega \in \mathbb{R} \tag{37}$$

where $\hat{v}(j\omega)$ and $\hat{w}(j\omega)$ are Fourier transforms of $v$ and $w$, respectively, $(\cdot)^*$ denotes the complex conjugate. Then, the closed-loop stability of the feedback interconnection can be verified by the Integral Quadratic Constraint (IQC) theorem (Megretski & Rantzer, 1997). Although the IQC framework allows for more general dynamic multipliers, here we only focus on the simple constant multiplier defined in (35).

**Theorem 3.** *Let $G$ be stable and $\phi$ be a static nonlinearity with sector bound of $[\alpha, \beta]$. The feedback interconnection of $G$ and $\phi$ is stable if here exists $\epsilon > 0$ such that*

$$\begin{bmatrix} G(j\omega) \\ I \end{bmatrix}^* \Pi \begin{bmatrix} G(j\omega) \\ I \end{bmatrix} \preceq -\epsilon I, \quad \forall \omega \in \mathbb{R}. \tag{38}$$

The Kalman-Yakubovich-Popov (KYP) lemma (Rantzer, 1996) can be applied to demonstrate the equivalence of Condition 3 in Theorem 3 to an LMI condition. The result is stated as follows.

**Theorem 4.** *There exists a $\epsilon > 0$ such that (38) holds if and only if there exists a matrix $P = P^\top$ such that*

$$\begin{bmatrix} A^\top P + PA & PB \\ B^\top P & 0 \end{bmatrix} + \begin{bmatrix} C^\top & 0 \\ 0 & I \end{bmatrix} \Pi \begin{bmatrix} C & 0 \\ 0 & I \end{bmatrix} \prec 0.$$

## D  LBEN PARAMETERIZATION FOR FEEDFORWARD NETWORKS

Given an equilibrium network (1) with weights $U, W$, and $W_o$, we can estimate its Lipschitz bound $\gamma$ by solving the following SDP with $(n + 1)$ decision variables:

$$\min_{\gamma > 0, \Lambda \in \mathbb{D}^+} \gamma \quad \text{s.t.} \quad \begin{bmatrix} 2\Lambda - \Lambda W - W^\top \Lambda & -\Lambda U & W_o^\top \\ -U^\top \Lambda & \gamma I & 0 \\ W_o & 0 & \gamma I \end{bmatrix} \succeq 0. \tag{39}$$

Note that the above LMI constraint is equivalent to (4) via Schur complement. A tight upper bound is then obtained by minimizing $\gamma$. When a deep neural network (a special case of equilibrium network) is considered, the above SDP yields the same bound estimation as **LipSDP-Neuron** in Fazlyab et al. (2019) since both formulations involve minimizing the gain bound $\gamma$ subject to an equivalent constraint (41).

Training a feedforward network with a prescribed Lipschitz bound is a challenge problem due to the LMI constraint (39) as well as the sparse structure of $W$. Following the similar idea of direct parameterization, we will construct a parameterization built on (9) to represent the following weight

$$W = \begin{bmatrix} 0 & & & \\ W_1 & \ddots & & \\ \vdots & \ddots & 0 & \\ 0 & \cdots & W_{L-1} & 0 \end{bmatrix}. \tag{40}$$

We first look at a simple case where $W$ is a dense strictly lower triangular matrix. Given a square matrix $H$, its LDU partition is defined as $H = [H]_D + [H]_L + [H]_U$ where $[H]_D$ is a diagonal matrix, $[H]_L([H]_U)$ is a strictly lower(upper) triangular matrix. Given any hyper-parameter $\gamma > 0$, the parameterization contains the following free variables: $V \in \mathbb{R}^{n \times n}, W_o \in \mathbb{R}^{p \times n}$, and $\widehat{U} \in \mathbb{R}^{n \times d}$. Let $S = [H]_L - [H]_L^\top, \Psi = [H]_D^{-1}$ and $U = \Psi \widehat{U}$ where $H = V^\top V + \epsilon I + (W_o^\top W_o + \widehat{U}\widehat{U}^\top)/2\gamma$. Then, the LBEN parameterization (9) yields

$$W = I - \Psi \left( \frac{1}{2\gamma} W_o^T W_o + \frac{1}{2\gamma} \Psi^{-1} U U^T \Psi^{-1} + V^T V + \epsilon I + S \right) = -2[H]_D^{-1}[H]_L,$$

which is a dense lower triangular matrix. To impose the sparse pattern like (40), we need

$$H = \begin{bmatrix} \Lambda_1 & H_1^\top & & & & \\ H_1 & \Lambda_2 & H_2^\top & & & \\ & H_2 & \Lambda 3 & H_3^\top & & \\ & & \ddots & \ddots & \ddots & \\ & & & H_{L-2} & \Lambda_{L-1} & H_{L-1}^\top \\ & & & & H_{L-1} & \Lambda_L \end{bmatrix}$$

where $\Lambda_i$ belongs to $\mathbb{D}^+$ with $1 \le i \le L$, and $H_j$ has the same dimension as $W_j$ for $1 \le j \le L-1$. To make $V^\top V$ have the same band structure as $H$, we further parameterize $V$ as follows

$$V = \begin{bmatrix} \Gamma_1 & & & \\ \Phi_1 V_1 & \Gamma_2 & & \\ & \ddots & \ddots & \\ & & \Phi_{L-1} V_{L-1} & \Gamma_L \end{bmatrix}$$

where $\Gamma_i, \Phi_j \in \mathbb{D}^+$ and $V_j^\top V_j = I$. The unitary matrix $V_j$ can be parameterized by $V_j = e^{S_j}$ where $S_j^\top = -S_j$. The diagonal blocks of $V^\top V$ are $\Gamma_i^2 + \Phi_i^2$ with $\Phi_L = 0$ while the lower off-diagonal blocks are $\Gamma_{j+1} \Phi_j V_j$ with $1 \le j \le L-1$. Similar techniques can be applied to the parameterization of $W_o$ and $\widehat{U}$.

# E PROOFS

## E.1 PROOF OF THEOREM 1

We presents two proofs for the well-posedness of equilibrium network (1). All these proofs are based on the following lemma.

**Lemma 1** (Simpson-Porco & Bullo (2014)). *For a time-invariant contracting dynamical system, all its solutions converge to a unique equilibrium.*

(*Monotone operator perspective*): This proof is mainly based on Proposition 2, which states that the solution of (1) is also a zero of the operator splitting problem $0 \in (A + B)(z)$, where the operators

$A$ and $B$ are given in (10). Condition 1 implies that the operator $A$ is strongly monotone while Assumption 1 implies that the operator $B$ is maximal monotone. Furthermore, the Clay operator $C_A$ is contractive and $C_B$ is non-expansive. Thus, applying Peaceman-Rachford algorithm to $0 \in (A + B)(z)$ yields a contracting discrete-time system (24) since $C_A C_B$ is a contractive operator. Since (24) is time-invariant, it yields a unique solution $z$ for any $x$ and $b_z$.

(*Neural ODE perspective*): This proof is built on Proposition 6, which states that the neural ODE (14) is a contracting continuous-time dynamical system under the Assumption 1 and Condition 1. For any fixed input $x$ and $b_z$, system (14) is also time-invariant and hence its solution converges to a unique equilibrium, which is also the solution of (1).

We now prove the Lipschitz boundedness of a well-posed equilibrium network. Condition 1 implies that there exists a constant $\epsilon > 0$ such that

$$2\Lambda - \Lambda W - W^T \Lambda \succeq \epsilon I.$$

For any $\delta \in (0, \epsilon)$ and weights $W_o, U$, we can find a sufficiently large but finite $\gamma$ such that

$$\frac{1}{\gamma}(W_o^T W_o + \Lambda U U^\top \Lambda) \preceq (\epsilon - \delta) I.$$

Then, Condition 2 holds for $\Lambda$ and $\gamma$ since

$$2\Lambda - \Lambda W - W^T \Lambda - \frac{1}{\gamma}(W_o^T W_o + \Lambda U U^\top \Lambda) \succeq \delta I \succ 0.$$

From Theorem 2, $\gamma$ is a Lipschitz bound for the well-posed equilibrium network (1).

## E.2 PROOF OF THEOREM 2

Rearranging Eq. (4) yields

$$2\Lambda - \Lambda W - W^T \Lambda \succ \frac{1}{\gamma}(W_o^T W_o + \Lambda U U^T \Lambda) \succeq 0.$$

The well-posedness of the equilibrium network (1) follows by Theorem 1. To obtain the Lipschitz bound, we first apply Schur complement to (4):

$$\begin{bmatrix} 2\Lambda - \Lambda W - W^\top \Lambda - \frac{1}{\gamma} W_o^\top W_o & -\Lambda U \\ -U^\top \Lambda & \gamma I \end{bmatrix} \succ 0.$$

Left-multiplying $\begin{bmatrix} \Delta_z^\top & \Delta_x^\top \end{bmatrix}$ and right-multiplying $\begin{bmatrix} \Delta_z^\top & \Delta_x^\top \end{bmatrix}^\top$ gives

$$2\Delta_z^\top \Lambda \Delta_z - 2\Delta_z^\top \Lambda W \Delta_z - \frac{1}{\gamma}\Delta_z^\top W_o^\top W_o \Delta_z - 2\Delta_z^\top \Lambda U \Delta_x + \gamma \|\Delta_x\|_2^2 \geq 0.$$

Since (5) implies $\Delta_v = W\Delta_z + U\Delta_x$ and $\Delta_y = W_o \Delta_z$, the above inequality is equivalent to

$$\gamma \|\Delta_x\|_2^2 - \frac{1}{\gamma}\|\Delta_y\|_2^2 \geq 2\Delta_z^\top \Lambda \Delta_z - 2\Delta_z \Lambda \Delta_v = 2\langle \Delta_v - \Delta_z, \Delta_z \rangle_\Lambda. \tag{41}$$

Then, the Lipschitz bound of $\gamma$ for the equilibrium network (1) follows by (6).

## E.3 PROOF OF PROPOSITION 1

(*if*): It is well-known that if $f$ is convex closed proper function, then $\mathbf{prox}_f^1$ is monotone and non-expansive, i.e., it is slope-restricted in $[0, 1]$. Here $f$ is not necessary to be closed as $\mathbf{dom}\, f$ (i.e. the range of $\sigma$) could be open interval $(z_l, z_r)$ or half-open interval $(z_l, z_r]$ or $[z_l, z_r)$. This can be resolved by defining $\hat{f}$ as the restriction of $f$ on the closed interval $[\hat{z}_l, \hat{z}_r]$, and then make $\hat{z}_l \to z_l$ and $\hat{z}_r \to z_r$.

(*only if*): Assumption 1 implies that $\sigma$ is a non-decreasing and piece-wise differentiable function on $\mathbb{R}$. Then, the range of $\sigma$ is an interval, denoted by $\mathcal{Z}$. We will construct the derivative function $f'$ on $\mathcal{Z}$ first and then integrate it to obtain $f$. Let $\{z_j \in \mathcal{Z}\}_{j \in \mathbb{Z}}$ be the sequence containing all points such that either $\sigma'(x_-) = 0$ or $\sigma'(x_+) = 0$ for all $x \in \sigma^{-1}(z_j)$. Note that $\sigma^{-1}(z)$ is a singleton for

| Activation | $\sigma(x)$ | Convex $f(z)$ | $\textbf{dom}\, f$ |
|---|---|---|---|
| ReLu | $\max(x, 0)$ | $0$ | $[0, \infty)$ |
| LeakyReLu | $\max(x, 0.01x)$ | $\frac{99}{2}\min(z,0)^2$ | $\mathbb{R}$ |
| Tanh | $\tanh(x)$ | $\frac{1}{2}\left[\ln(1-z^2) + z\ln\left(\frac{1+z}{1-z}\right) - z^2\right]$ | $(-1, 1)$ |
| Sigmoid | $1/(1+e^{-x})$ | $z\ln z + (1-z)\ln(1-z) - \frac{z^2}{2}$ | $(0, 1)$ |
| Arctan | $\arctan(x)$ | $-\ln(|\cos z|) - \frac{z^2}{2}$ | $(-1, 1)$ |
| Softplus | $\ln(1+e^x)$ | $-\text{Li}_2(e^z) - i\pi z - z^2/2$ | $(0, \infty)$ |

Table 3: A list of common activation functions $\sigma(x)$ and associated convex proper $f(z)$ whose proximal operator is $\sigma(x)$. For $z \notin \textbf{dom}\, f$, we have $f(z) = \infty$. In the case of Softplus activation, $\text{Li}_s(z)$ is the polylogarithm function.

all $z \in (z_j, z_{j+1})$, whereas $\sigma^{-1}(z_j)$ is a closed interval of the forms $(-\infty, x_r]$, $[x_l, x_r]$ or $[x_l, \infty)$. Then, we define $f'$ as follows

$$f'(z) = \begin{cases} \min[\sigma^{-1}(z)] - z, & \text{if } z = z_j \text{ and } \min \sigma^{-1}(z) > -\infty, \\ \max[\sigma^{-1}(z)] - z, & \text{if } z = z_j \text{ and } \min \sigma^{-1}(z) = -\infty, \\ \sigma^{-1}(z) - z, & \text{otherwise.} \end{cases}$$

Without loss of generality, we assume that $0 \in \mathcal{Z}$ and $\sigma^{-1}(0)$ is well-defined. We define the function $f$ as follows

$$f(z) = \begin{cases} \int_0^z f'(\zeta)d\zeta + C & \text{if } z \in \mathcal{Z}, \\ \infty & \text{otherwise,} \end{cases}$$

where $C$ is an arbitrary constant. Note that $f$ is a convex function as $f'$ is a piecewise differentiable function on $\mathcal{Z}$ and for those points where $x = \sigma^{-1}(z)$ is well-defined, $f'$ is differentiable with $f''(z) = 1/\sigma'(x) - 1 \geq 0$ as $\sigma'(x) \in (0, 1]$. Finally, the definition of $f'$ implies that $0 \in z - \sigma^{-1}(z) + \partial f(z)$, which implies that $z = \sigma(x)$ is the unique minimizer of $1/2(z - x)^2 + f(z)$. Furthermore, since $\sigma$ is well-defined, we can conclude that $f$ is bounded from below. We also provide a list of $f$ for common activation functions in Table 3. A similar list can also be found in Li et al. (2019).

### E.4 PROOF OF PROPOSITION 2

Similar to Winston & Kolter (2020), we first show that the solution of (1), if it exists, is an fixed point of the forward-backward iteration (23) with $\alpha = 1$:

$$z^{k+1} = R_B(z^k - \alpha A z^k) = \textbf{prox}_f^1(z^k - \alpha(I - W)z^k + \alpha(Ux + b_z)) = \sigma(Wz^k + Ux + b_z).$$

The last equality follows by

$$\sigma(x) = \begin{bmatrix} \arg\min_{z_1} \frac{1}{2}(z_1 - x_1)^2 + f(z_1) \\ \vdots \\ \arg\min_{z_n} \frac{1}{2}(z_n - x_n)^2 + f(z_n) \end{bmatrix} = \arg\min_z \frac{1}{2}\|z - x\|_\Lambda^2 + \sum_{i=1}^n \lambda_i f(z_i) = \textbf{prox}_f^1(x).$$

Note that the necessary condition for $\sigma(\cdot)$ to be diagonal is that the weight $\Lambda$ is positive diagonal.

Now we prove the well-posedness of LBEN by showing that the operator splitting problem $0 \in (A + B)(z)$ has a unique solution for any $x$ and $b_z$. Both Condition 1 and 2 implies that the operator $A$ is strongly monotone and its Cayley operator $C_A$ is contractive. Then, the Peaceman-Rachford iteration (24) is contracting and hence it converges to a unique fixed point.

### E.5 Proof of Proposition 3

The matrix $J$ is diagonal with elements in $[0, 1]$. Decompose $\Lambda = \Pi(J + \mu I)$ for some small $\mu > 0$, i.e. $\Pi = \Lambda(J + \mu I)^{-1}$, which is diagonal and positive-definite. By denoting $H = \Pi(I - W) + (I - W)^T \Pi$ we obtain the following inequality from (3):

$$\Pi J(I - W) + (I - W)^T J \Pi + \mu H \succeq \epsilon I,$$

which can be rearranged as

$$\Pi(I - JW) + (I - JW)^T \Pi \succeq \epsilon I + 2\Pi(I - J) - \mu H.$$

Since $2\Pi(I - J) \succeq 0$, we can choose a sufficiently small $\mu$ such that

$$\Pi(I - JW) + (I - JW)^T \Pi \succ 0,$$

which further implies that $I - JW$ is strongly monotone w.r.t. $\Pi$-weighted inner product, and is therefore invertible.

### E.6 Proof of Proposition 4

First, we show that (12) is strongly convex. Since $\mathfrak{f}(z)$ is a conic combination of convex functions $f(z_i)$, we only need to show that the quadratic term is strongly convex, i.e.,

$$\nabla^2 J = \Lambda(I - W) + (I - W)^\top \Lambda \succ 0$$

where

$$J(z) = \left\langle \frac{1}{2}(I - W)z - Ux - b_z, z \right\rangle_\Lambda$$

which follows by either Condition 1 or (2). Moreover, since $S = 0$ for the direction parameterization of $W$, we have $\Lambda(I - W) = (I - W)^\top \Lambda$ and hence $\partial J = A$. Then, finding the global minimizer of the strongly convex optimization problem (12) is equivalent to finding a zero for the operator splitting problem $0 \in \partial(J + \mathfrak{f})(z) = (A + B)(z)$.

### E.7 Proof of Proposition 5

The proof is based on the "key insights" of ReLU activation from Raghunathan et al. (2018b). That is, a ReLU constraint $z = \max(x, 0)$ is equivalent to the following three linear and quadratic constraints between $z$ and $x$: (i) $z(z - x) = 0$, (ii) $z \geq x$, and (iii) $z \geq 0$. From this observation an equilibrium network (1) can be equivalently expressed as the following constraints (I) $z^\top(z - q) = 0$, (II) $z \geq q$, and (III) $z \geq 0$, where $q = Wz + Ux + b$. Note that (II) and (III) can be rewritten as the linear constraints in the QP problem (13) while (I) is equivalent to $J(z) = 0$ with

$$J(z) := z^\top \Lambda(z - q) = \frac{1}{2}z^\top Hz + p^\top z$$

for any $\Lambda \in \mathbb{D}^+$. It is obvious that $J(z) \geq 0$ for all $z$ satisfying (II) and (III), and hence the solution of (1) is a global minimizer of the QP problem (13). If $\Lambda$ satisfies either Condition 1 or 2, then $H$ is positive-definite and(13) is a strongly convex QP problem. Thus, its global minimizer is unique, which is also the solution of LBEN (1).

### E.8 Proof of Proposition 6

From (14) the dynamics of $\Delta_v$ and $\Delta_z$ can be formulated as a feedback interconnection of a linear system $\dot{\Delta}_v = -\Delta_v + W\Delta_z$ and a static nonlinearity $\Delta_z = \sigma(v_a) - \sigma(v_b)$. The linear system can be represented by a transfer function is $G(s) = 1/(s + 1)W$. The nonlinear component can be rewritten as $\Delta_z = \Phi(v_a, v_b)\Delta_v$ where $\Phi$ as a diagonal matrix with each $\Phi_{ii} \in [0, 1]$. For the nonlinear component $\Phi$, its input and output signals satisfies the quadratic constraint (6). For the linear system $G$, we have the following lemma.

**Lemma 2.** *If Condition 1 holds, then for all $\omega \in \{\mathbb{R} \cup \infty\}$*

$$\begin{bmatrix} G(j\omega) \\ I \end{bmatrix}^* \begin{bmatrix} 0 & \Lambda \\ \Lambda & -2\Lambda \end{bmatrix} \begin{bmatrix} G(j\omega) \\ I \end{bmatrix} \prec 0. \tag{42}$$

The KYP Lemma (Theorem 4) states that (42) is equivalent to the existence of a $P = P^\top$ such that

$$\begin{bmatrix} -2P & PW \\ W^T P & 0 \end{bmatrix} + \begin{bmatrix} 0 & \Lambda \\ \Lambda & -2\Lambda \end{bmatrix} \prec 0.$$

It is clear from the upper-left block that $P \succ 0$. The above inequality also implies

$$2\langle -\Delta_v + W\Delta_z, \Delta_v \rangle_P \leq \langle \Delta_z - \Delta_v, \Delta_z \rangle_\Lambda - \epsilon(\|\Delta_z\|_2^2 + \|\Delta_v\|_2^2) \leq -\epsilon(\|\Delta_z\|_2^2 + \|\Delta_v\|_2^2)$$

for some $\epsilon > 0$. The contraction property of the neural ODE (14 follows since

$$\frac{d}{dt}\|\Delta_v\|_P^2 = 2\langle -\Delta_v + W\Delta_z, \Delta_v \rangle_P \leq -\epsilon(\|\Delta_z\|_2^2 + \|\Delta_v\|_2^2) \leq -2\beta\|\Delta_v\|_P^2$$

for some sufficiently small $\beta > 0$. As a byproduct of the above inequality, we will show that the operator $-f$ with with $f(v) = -v + W\sigma(v) + Ux + b_z$ is strictly monotone w.r.t. the $P$-weighted inner product since

$$\langle -f(v_a) + f(v_b), v_a - v_b \rangle_P = \langle \Delta_v - W\Delta_z, \Delta_v \rangle_P \geq \beta\|\Delta_v\|_P^2.$$

### E.9 PROOF OF LEMMA 2

Note that (42) is equivalent to

$$2\Lambda - G_0(j\omega)\Lambda W - G_0(-j\omega)W^T\Lambda \succeq \mu I \tag{43}$$

where $G_0(j\omega) = \frac{1}{1+j\omega}$. For some $\omega \in (\mathbb{R} \cup \infty)$ let $g = \Re G_0(j\omega) = \Re G_0(-j\omega)$, where $\Re$ denotes real part. It is easy to verify that $g = 1/(\omega^2 + 1) \in [0,1]$. From (3) we have

$$2g\Lambda - g\Lambda W - gW^T\Lambda \succeq g\epsilon I$$

for some $\epsilon > 0$. Rearranging the above inequality yields

$$2\Lambda - g\Lambda W - gW^T\Lambda \succeq g\epsilon I + (1-g)2\Lambda$$

Now, since $g \in [0,1]$ the right-hand-side is a convex combination of two positive definite matrices: $\epsilon I$ and $2\Lambda$, therefore (43) holds for some $\mu > 0$ and all $\omega \in (\mathbb{R} \cup \infty)$.

### E.10 PROOF OF PROPOSITION 7

It is straightforward to verify that an equilibrium network with the following weights is identical to the feedforward network (15):

$$z = \begin{bmatrix} z_1 \\ z_2 \\ \vdots \\ z_L \end{bmatrix}, \quad W = \begin{bmatrix} 0 & & & \\ W_1 & \ddots & & \\ \vdots & \ddots & 0 & \\ 0 & \cdots & W_{L-1} & 0 \end{bmatrix}, \quad U = \begin{bmatrix} U_0 \\ 0 \\ \vdots \\ 0 \end{bmatrix}, \quad W_o = \begin{bmatrix} 0 & \cdots & 0 & W_L \end{bmatrix}. \tag{44}$$

To construct an LBEN parameterization in the form (8) for $W$, we first need the following lemma.

**Lemma 3.** *Condition 1 holds for any strictly lower triangular $W$.*

*Proof.* We prove it by showing that for any $\delta > 0$, there exists a $\Lambda \in \mathbb{D}^+$ such that

$$H(\Lambda_n, W_n) := \Lambda_n(I - W_n) + (I - W_n)^\top \Lambda_n \succ 2^{2-n}\delta I. \tag{45}$$

where $\Lambda_n, W_n$ are the upper left $n \times n$ elements of $\Lambda, W$, respectively. For $n = 1$, $\lambda_1 > \delta$ is sufficient since $W_1 = 0$. Assuming that (45) holds for $\Lambda_n$ and $W_n$, then we have

$$H(\Lambda_{n+1}, W_{n+1}) - 2^{1-n}\delta I = \begin{bmatrix} H(\Lambda_n, W_n) - 2^{1-n}\delta I & -\Lambda_n w_{n+1}^\top \\ -w_{n+1}\Lambda_n & 2(\lambda_{n+1} - 2^{-n}\delta) \end{bmatrix}, \tag{46}$$

where $\Lambda_{n+1} = \text{diag}(\Lambda_n, \lambda_{n+1})$ and $W_{n+1} = \begin{bmatrix} [W_n \ 0] & 0 \\ w_{n+1} & 0 \end{bmatrix}$. By applying Schur complement to (46), Inequality (45) holds for the case of $n+1$ if $\lambda_{n+1} > 2^{-n}\delta + 2^{n-2}|\Lambda_n w_{n+1}|^2/\delta$. $\square$

Based on the above lemma, we can construct a $V$ such that $V^\top V = 1/2[\Lambda(I-W)+(I-W)^\top\Lambda]-\epsilon I$ where $\epsilon = 2^{1-n}\delta$. By choosing $\Psi = \Lambda^{-1}$ and $S = (\Lambda W - W^\top \Lambda)/2$, the LBEN parameterization (8) recovers the exact $W$. Thus, LBEN contains all feedforward networks (44).

We note that "skip connections" as in a residual network can easily be added to the above structure via additional non-zero blocks in the lower-left part of the weight $W$.

### E.11 PROOF OF PROPOSITION 8

From the MON parameterization (16) we have

$$H(m, W) := 2(1 - m)I - W - W^\top = 2A^\top A \succeq 0.$$

Let $\mathcal{W}_m$ be the set of non-zero and strictly lower triangular $W$ such that $H(m, W) \succeq 0$. Note that $\mathcal{W}_{m_1} \subset \mathcal{W}_{m_2}$ if $m_1 > m_2$. Because $H(m_1, W) \succeq 0$ implies $H(m_2, W) = H(m_1, W) + 2(m_1 - m_2)I \succ 0$ for all $m_2 < m_1$. Proposition 8 follows if $\lim_{m \to 0} \mathcal{W}_m$ does not contain all strictly lower triangular $W$. Since $W$ is a strictly lower triangular, $H(0, W)$ is a semidefinite matrix whose diagnoal elements are 2. As the norm of $W$ increases, $H(0, W)$ becomes indefinite. Taking the feedforward network (44) with $L = 2$ as an example, the set of $\mathcal{W}_0$ is characterized by $W_1 W_1^\top \preceq 4I$ since

$$H(0, W) = \begin{bmatrix} 2I & -W_1^\top \\ -W_1 & 2I \end{bmatrix} \succeq 0.$$

Now we show that $\mathcal{W}_m = \emptyset$ for all $m \geq 1$. Since the diagnoal elements of $H(m, W)$ are non-positive when $m \geq 1$, the matrix $H(m, W)$ is not semi-definite for any strictly lower triangular $W$.

## F  TRAINING DETAILS

### F.1  MNIST EXAMPLE

This section contains the model structures and the details of the training procedure used for the MNIST examples. All models are trained using the ADAM optimizer Kingma & Ba (2015) with an initial learning rate of $1 \times 10^3$. All models are trained for 40 Epochs, and the learning rate is reduced by a factor of 10 every 10 epochs.

The models in the MNIST example are all fully connected models with 80 hidden neurons and ReLU activations. For the equilibrium models, the forward and backward passes models are performed using the Peaceman-Rachford iteration scheme with $\epsilon = 1$ and a tolerance of $1 \times 10^{-2}$. When evaluating the models, we decrease the tolerance of the spitting method to $1 \times 10^{-4}$. We use the same $\alpha$ tuning procedure as Winston & Kolter (2020). All models were trained using the same initial point. Note that for LBEN, this requires initializing the metric $\Lambda = I$.

The feed-forward models trained using Lipschitz margin training were trained using the original author's code which can be found at `https://github.com/ytsmiling/lmt`.

### F.2  CIFAR-10 EXAMPLE

This section contains the model structures and the details of the training procedure used for the CIFAR-10 examples. All models are trained using the ADAM optimizer Kingma & Ba (2015) with an initial learning rate of $1 \times 10^3$. The models were trained for 25 epochs and the learning rate was reduced by a factor of 10 after 15 epochs. Each model contains a single convolutional layer, an average pooling layer with kernel size 2, and a linear output layer.

The convolutional LBEN has 81 channels and is parametrized as discussed below. The MON similarly has 81 channels. Unless otherwise stated, the feed-forward convolutional network has 162 channels which gives it approximately the same number of parameters as the LBEN.

The MON was evaluated using the Peaceman-Rachford Iteration scheme.

#### CONVOLUTIONAL LBEN

Following the approach of Winston & Kolter (2020), we parametrize $U$ and $V$ in equation 9 via convolutions. The skew symmetric matrix is constructed by taking the skew symmetric part of a convolution $\bar{S}$, so that $S = \frac{1}{2}(\bar{S} - \bar{S}^\top)$. Similar, to Winston & Kolter (2020), we also find that using a weight normalized parametrization improves performance. Specifically, we use the following parametrization: $V = \sqrt{\alpha} \frac{\hat{V}}{|\hat{V}|}$, $\bar{S} = \beta \frac{\hat{S}}{|\hat{S}|}$, $U = \sqrt{\eta} \frac{\hat{U}}{|\hat{U}|}$ and $W_o = \sqrt{\xi} \frac{\hat{W}_o}{|\hat{W}_o|}$.

In Winston & Kolter (2020) Peaceman-Rachford is used and the operator $I - W$ can be quickly inverted using the fast Fourier transform. This situation is more complicated in our case as the term $W_{\text{out}}^{\top} W_{\text{out}}$ cannot be represented as a strict convolution and this is not diagonalized by the Fourier matrix,. Instead, we apply Forward-Backward Splitting algorithm shown in equation 23 which does not require a matrix inversion.

We have observed that the rate of convergence of the Forward-Backward splitting algorithm is highly dependent on the monotonicity parameter $m$. In particular, for the convolutional models, we found there was a strong trade-off between the ease of solve for the equilibrium versus the model expressibility and the accuracy of the Lipschitz bound.

