# OpenReview forum: "Lipschitz-Bounded Equilibrium Networks"
_ICLR.cc/2021/Conference — Reject_

### Official Review · AnonReviewer1 · 2020-10-26
**Solid theoretical results that generalized previous formulations; empirical results could be improved**

**Rating:** 7
**Confidence:** 4

**Review:**

> Summary: This paper studies a new and more general way of parameterizing the simplest equilibrium network of the form $\sigma(Wz+Ux+b)$, a form that has been tackled by works like (Winston & Kolter 2020)and (El Ghaoui et al. 2019). The authors provide a computationally (relatively) efficient way of computing Lipschitz-bounded equilibrium networks and a detailed analysis of how the network should be constructed, along with the proof of the existence and uniqueness of the fixed point (and with less restrictive conditions when compared to MON). The empirical results on adversarial robustness shows that the proposed approach is a bit more robust than prior layer-based networks and other implicit networks, and validates most of the theoretical claims made by the authors.

My general opinion is that this paper extends the discussion made in (Winston & Kolter 2020) with its Condition 1 and 2 (e.g., they show that MON is simply a special case of condition 1 with $\Lambda=I$), and thus provides a reasonable method for constructing Lipschitz bounded equilibrium networks. The paper is overall well-written, and the insights are explained in a clear fashion (I especially like the Example 1, where the authors showed that Condition 1 is a less restrictive formulation). The experimental results validates the theoretical claims of the authors by demonstrating the effectiveness of the approach, but they could also be made stronger. But in general, I think this paper offers an insightful extension on the prior work(s) in this direction, and is an important addition to the implicit-depth neural network literature.

Pros:

1. Well-written paper with clearly defined notations and dimensionalities. The proofs are generally easy to follow (and pretty standard, to be honest).
2. The theoretical and empirical results demonstrate the improvement over the previous MON approach.
3. The authors additionally study the relationship between LBEN and neural ODEs, which to my knowledge is new (in terms of formally connecting these two implicit models).

Cons:

1. The empirical results are still on very small scales and not quite convincing.
2. While the relaxation of the constraints is novel, many other theoretical contents were already discussed in the MON paper. For example, the non-singularity of $I-JW$, the monotonicity and non-expansiveness of the proximal operators (a well-known result), Appendix B, etc. Granted, the two papers also have slightly different focuses (e.g., MON talked more about the PR algorithm).
3. The scope of the paper is still on the simplest $\sigma(Wz+Ux+b)$ form, which has been studied before. There is a lack of discussion on the challenge involved in building Lipschitz equilibrium models more generalized than this, and it's unclear how well some of the claims of the authors hold in a general setting (see my questions below).

---------------------------------

Some more questions/comments:

1. I didn't find the proof to Theorem 1's claim on "finite Lipschitz bound from $x$ to $y$" in the paper. I didn't find a formal "proof" to Theorem 1, and the proof of uniqueness is simply on page 4. But what does Assumption 1 and Condition 1 imply about the Lipschitz constant?

2. In the Neural ODE discussion, the paper claims that "for any well-posed equilibrium network, there corresponds a contracting (strongly stable) neural ODE", which intuitively makes sense to me (as one can think of a deep equilibrium model as a backward Euler...?). But more strictly speaking, is this supposed to be a general statement for deep equilibrium neural networks, or just for networks of the form (1) in the paper (i.e., $\sigma (Wz+Uz+b)$)? If the latter, I think the authors may want to clarify this; if the former, I think a proof would be hard (e.g., the (Bai et al. 2019) DEQ-Transformers), but I'd appreciate if the authors can give more theoretical insights on this.

3. One concerning part is the empirical study of the approach. To be clear, I think the authors did a good job validating the approach by comparing with MON and unconstrained DEQs (where their finding seem to agree with prior works that these unconstrained DEQs eventually have pretty bad Lipschitz constant). But I think two important pieces are missing. First, since Condition 1 (and thus Eq. (10) in the paper) provide a less restrictive formulation than MON, one would naturally think that the LBEN built in this way could have stronger representational power. But the results do *not* seem to suggest so. The $\text{LBEN}_{\gamma < \infty}$ seems overall very close to the $\text{MON}$ results in Table 1 and Figure 3(a). Second, (and together with the first point), I think it would be more useful to study a larger dataset, like CIFAR-10/100 (which MON did provide), with a convolutional LBEN. I'm curious to see how it compares with MON and other implicit models there.

4. Since LBEN also uses similar method(s) as MON to compute the equilibrium point (e.g., Peaceman-Rachford), I'm assuming this method is not "more scalable" than the previous methods as you are solving the same high-dimensional equilibrium system? Can the authors confirm this? What about the memory efficiency during training?

---------------------------------

Some minor things:

- i) I didn't realize that the text following Theorem 2 on page 4 is the proof to Theorem 1 & 2 (as there is a "but first we make some straightforward remarks" sentence). It'd also be helpful, I think, to say something like you are assuming for (the sake of) contradiction that there are two solutions $z_a \neq z_b$. It was a bit surprising to me to see $z_a$ and $z_b$ as the theorem claimed "uniqueness".

- ii) Page 4 "on the other size" ---> "on the other side"?

---

> ### Author Response · Authors · 2020-11-24
> **Thank you for your comments, the revised paper will include improved empirical results**
>
> We thank the reviewer for their careful reading, positive comments, and constructive feedback.
>
> Here we give some brief responses to the "Cons" and the reviewer's questions.
>
> Cons:
> 1. As mentioned in the response to the first reviewer, we agree that the submitted paper had quite rudimentary empirical results. Since receiving the reviews, we have been able to generate results with larger convolutional networks with 81 channels on the CIFAR10 dataset. While these are arguably still not "large-scale", we have found that in fact we can still achieve both (slightly) improved nominal performance and (dramatically) improved robustness over MON, unconstrained, and feedforward networks.  These will be presented in the revised version.
> 2. We agree that many of the theoretical results in the appendices parallel those presented in Winston & Kolter. However, that paper worked entirely in a Euclidean space, whereas we work with non-Euclidean inner-products, so for completeness we have included the relevant results for this case in the appendices. Note these are not presented as novel results of the paper, simply as required background for our main results, and most can be found in standard textbooks such as Baushke et al, Convex analysis and monotone operator theory in Hilbert spaces.
> 3. Roughly speaking, the challenge in extending our approach to equilibrium networks with more general structure is the same as extending robustness and contraction analysis of dynamical systems to more general dynamical equations. In the simple structure, $\sigma(Wz+Ux+b)$ we have taken advantage of the fact that this decomposes into feedback connection of a linear operator and a simple monotone nonlinearity, which corresponds to a well-studied class of dynamical systems (Lur\'e systems). We have also mentioned in the conclusion (with a reference) that the same approach can be extended to certain other classes of nonlinearities, e.g. when the ``activation function'' is the arg-min of a class of quadratic programs.
>
> In general, the most powerful computational techniques for robustness analysis of nonlinear systems, including integral quadratic constraints and sum-of-squares, are based around finding some (non-convex) quadratic representation of the space of possible behaviours, and then verifying robustness via semidefinite programming. This approach is relatively straightforward to extend to sequence modelling via recurrent neural networks and LSTM. However, in the case of DEQ-Transformers it is not immediately obvious to the authors how the attention mechanism could be accurately represented in this framework.
>
>
>
> Questions/comments:
> 1. Thank you for this comment, we agree that this was not clear in the original submission. In the revised version we will give a formal proof of Theorem 1 in an appendix.
> 2. We agree that this was unclear in the original paper. Perhaps a better way to state it is that any monotone equilibrium network (with respect to any metric) corresponds to a contracting neural ODE. This is a consequence of the fact that the mathematical definition of a contracting vector field is actually identical to the definition of a monotone operator (with a sign change). In a broader sense, it can be possible that a dynamical system has a unique equilibrium for all inputs without being a contraction: the necessary condition is the existence of a Lyapunov function for each equilibrium, which is a weaker condition. However, a Lyapunov function must be constructed about a known equilibrium, so it is not usually possible to apply during training, since the location of the equilibrium will depend on the inputs and weight matrices.
> 3. Thank you for this detailed comment. As mentioned above,  since the reviews were released we have generated new results on the
> CIFAR10 data set. We do indeed see that the new parameterisation has stronger representative power, although it must be said that the effect is often subtle. In our view the more significant result of the paper is the ability to impose Lipschitz bounds directly.  The improved results will be presented in the revised paper. In addition, we have also generated some new theoretical results related to expressivity; the LBEN parameterisation includes all possible feedforward networks, whereas MON only includes feedforward networks with a particular bound on the weight matrix, and in fact using the parameters that were used by Winston & Kolter, MON does not include any feedforward networks within its model class.
> 4. This is correct, in terms of scalability and efficiency during training, LBEN is essentially identical to MON. The main benefit is the ability to impose Lipschitz bounds.

---

> ### Comment · AnonReviewer1 · 2020-11-24
> **Thank you for the response**
>
> I have read the authors' responses and am generally satisfied with the answers. Thank you!
>
> This paper provides a clear theoretical analysis of the Lipschitz properties of (a simple version of) Deep Equilibrium Models, while also drawing a connection with other implicit models (e.g., Neural ODEs).
>
> I look forward to reading the revised paper (it seems it's still not updated on OpenReview; I'd strongly recommend that you upload by the author response deadline. The original empirical results seem to be a (slight) concern shared by multiple reviewers.)!

---

### Official Review · AnonReviewer4 · 2020-10-27
**A direct parameterization for Lipschitz-bounded equilibrium networks**

**Rating:** 6
**Confidence:** 2

**Review:**

The paper provides a novel approach to guarantee Lipschitz for a special type of equilibrium network. Since I am not an expert in equilibrium networks, I can only comment on high-level problems.

(a) The use of equilibrium networks is not well motivated. Different from a feed-forward network, the computation of an equilibrium network is not always well-posed. It raises the question of why the equilibrium is adopted at the very beginning. While equilibrium networks generalize traditional networks, it is not well explained why people need to deal with the complicated constraints instead of using a traditional network. The toy problems in the experiments are relatively easier to solve using traditional networks.

(b) The choice of equilibrium equation (1) is not very well explained. The fact that it covers a deep or residual network as a special case is not a good reason for generalization --- we hope the generalization has additional properties that a special case does not have. It seems the paper only addresses the direct parameterization of the particular choice of equation (1) and can not be adopted if the equilibrium equation is changed.

I was concerned about the method because the proposed method seems to overkill the original version's simple experiments. The newly added experiments and clarifications make sense to me.

---

> ### Author Response · Authors · 2020-11-24
> **Thank you for your comments, the revised paper will better motivate the use of this model class**
>
> We thank the reviewer for careful reading and insightful comments.
>
> Regarding point (a) we agree that the motivation can be improved, and we are working on this for the revised paper. To make some brief comments here:
> 1. On well-posedness: It is true in general that an equilibrium network is not always well-posed, but for our proposed class of networks it the computation is always well-posed.
> 2. On "complicated constraints": In fact, our method does not require any constraints. While our conditions are first presented as constraints, arguably the central contributions of our paper is a direct parameterization for which well-posedness and, if desired, a specific Lipschitz bound are be guaranteed by construction without requiring any complicated constraints during learning.
> 3. "The toy problems in the experiments are relatively easier to solve using traditional networks.": This may be true, depending on what is meant by "solve". During the rebuttal period we have been able to generate results comparing LBEN to traditional feedforward networks of the same size on CIFAR10, and we observe that our parameterisation results in significantly better nominal performance, lower observed Lipschitz bound, and dramatically better emprical robustness to adversarial attack.
>
> Regarding point (b) as mentioned above, compared to feedforward structures our parameterisation can result in both better nominal performance and better robustness. We will work to improve the presentation of these results in the revised paper.
>
> It is true that our paper only considers the paramterization (1), as was considered in both Winston & Kolter and El Ghaoui et al. Extending to  broader classes of equilibrium networks is an interesting task, but it is not trivial to give results that are both general and computationally tractable. In the authors view, this is a major program of research and far more than can be achieved in a single paper. Indeed, the problem is essentially equivalent to computationally verifying existence and robust stability of equilibria of arbitrary nonlinear dynamical systems, a problem which has been the focus of sustained research in control theory for more than five decades, and still is far from solved.

---

### Official Review · AnonReviewer2 · 2020-10-27
**new parameterization of Lipschitz deep equilibrium model**

**Rating:** 6
**Confidence:** 3

**Review:**


Summary: The paper introduces a new condition for showing the existence of the solution of a deep equilibrium model (which defines an implicit mapping via the fixed point). The new formulation also comes with a convenient and accurate Lipschitz bound. The proposed condition can be satisfied via reparameterizing an unconstrained set of trainable parameters.


Contributions: introduces a new parameterization of DEM. The parameterization satisfies a condition which is shown to imply the existence of the unique solution of the fixed point. The paper introduces two proofs: the solution of the DEM can be shown to be equivalent to 1) the minimizer of a strongly convex potential and 2) the equilibrium of a contractive neural ODE. The results seem nontrivial (although I did not check the correctness).


Weakness: the paper is more theoretical, and weaker on the empirical side. The new condition on the weights are less restrictive than the condition of Winston & Kolter, which means the model has better expressivity. However, the benefit of this extra expressivity is not discussed much and demonstrated empirically, which is why I lean towards weak rejection. The model used in the experiment section is also very small (80 hidden units for MNIST), which does not seem very realistic for testing adversarial attack.

Also, I think the paper is quite borderline given its scope, and it depends on the wider adoptability of DEM. It will help me better understand its potential impact if the authors could try to motivate it better, explaining the need to further study the theoretical properties of DEM.

Additional details
- What is the L2 fast gradient sign method? FGSM restricts the L infinity norm of the perturbation.
- Please provide some visualization of the perturbed and unperturbed data points.


Minor points:
- The discussion of computational complexity can be moved to the main text (right now it’s not really in Fig 4, it’s in its caption). Also are all four variants run on the same machine? Why is unconstrained slower?
- Typo: between (7) and (8), “on the other size*”


=== AFTER REBUTTAL ===

The authors have raised a few fair points in the rebuttal (especially point 1), so I've adjusted my rating accordingly.

---

> ### Author Response · Authors · 2020-11-24
> **Thank you for your comments, in the revised version we will show improved empirical results illustrating the benefits of our method**
>
> Thank you for your comment, we agree that the main strength of this paper is its theoretical content.
>
> I'm not sure if it was clear, but we note that our new representation not only offers more expressivity than Winston & Kolter, but also the ability to directly impose Lipschitz bounds during training. In fact, it is the latter that we feel is the more significant contribution.
>
> As mentioned in the response to the previous reviewer, we agree that the submitted paper had quite rudimentary empirical results. Since receiving the reviews, we have been able to generate results with larger convolutional networks with 81 channels on the CIFAR10 dataset. While these are arguably still not "large-scale", we have found that in fact we can achieve both (slightly) improved nominal performance and (dramatically) improved robustness over MON, unconstrained, and feedforward networks.
>
> Regarding the scope, while we agree to some extent with the reviewer that the impact of our work will be strongly tied to the wider adoptability of equilibrium models, we believe there are some caveats to this argument:
>
> 1. Our paper does not only "further study the theoretical properties of DEM", it introduces a new class of DEM with guaranteed robustness properties. So in a sense there is a ``chicken and egg' problem behind this argument. The main claim of our paper is that it presents improvements to equilibrium networks, in particular the ability to add robustness guarantees without extra computational complexity. The argument that improvements to a method should not be published until that method finds wide adoptability is problematic since it may be improvements such as those we propose (most likely when combined by many other improvements by other researchers) that leads to wide adoptability.
> 2. Our results are not limited to Equilibrium models. In fact, our paper also presents results which are novel for standard feedforward networks, which have unambiguously found wide applicability. In particular, our results allow imposing Lipschitz bounds that are as tight as the best-known bounds, but without adding any extra complexity during training. We admit that the original version of the paper did not make this point clearly, and we are working on improving this substantially in the revised version. When compared to the only other method we are aware of that can impose the same class of bounds during training (Pauli et al, arXiv:2005.02929, 2020) our method is orders of magnitude faster.
>
> On the additional details, The L2 fast gradient method is an adversarial attack implemented as part of the foolbox toolbox. It is similar to the FGSM, except it restricts the L2 norm of the perturbation rather than L-infinity.
>
> As requested, we will add to the revised paper visualizations of example perturbed and unperturbed data points.

---

### Official Review · AnonReviewer3 · 2020-10-28
**Interesting Idea, Very Compelling Theory and Relatively Weak Experiments**

**Rating:** 8
**Confidence:** 4

**Review:**

Objective: Find a way to parametrize equilibrium neural networks to allow setting a Lipschitz bound on the input-output mapping.

Central claims:
1.	There exists an extension to the parametrization proposed in Monotone Operator Equilibrium Networks (MON) that allows for explicitly setting a Lipschitz bound.
2.	The resulting networks (dubbed LBEN here) can be practically trained via. unconstrained optimization, which involves computing the equilibrium using tools from convex optimization.
3.	Empirically, LBEN models can achieve comparable performance with Monotone Operator Equilibrium Networks, their pre-set Lipschitz constants are tight and therefore achieve favorable accuracy-robustness tradeoff.

Strong points:
1.	Interesting idea with very strong theoretical backing: I found the theoretical development and analysis of LBEN models very rich and interesting. There are a bunch of novel results scattered throughout, and the analysis draws from a wide variety of disciplines, including convex optimization, dynamical systems theory and control theory. I found the paper delightful to read. The properties that LBEN models are proven to possess (well-posedness under less restrictive conditions than MON and more natural assumptions on the activation functions) are quite compelling. LBEN models also don’t have any extra computational overhad over MON models.
2.	Claims are well supported. Claims 1 and 2 are very well supported in theory. Claim 3 can only be confirmed for very small scale experiments.


Weak points: I believe the only relative weakness of the paper is its experiments section.
1.	Lack of large scale experiments: The models trained in the experiments section are quite small (80 hidden neurons for the MNIST experiments and a single convolutional layer with 40 channels for the SVHN experiments). It would be nice if there were at least some experiments that varied the size of the network and showed a trend indicating that the results from the small-scale experiments will (or will not) extend to larger scale experiments.
2.	Need for more robustness benchmarks: It is impressive that the Lipschitz constraints achieved by LBEN appear to be tight. Given this, it would be interesting to see how LBEN’s accuracy-robustness tradeoff compare with other architectures designed to have tight Lipschitz constraints, such as [1].
3.	Possibly limited applicability to more structured layers like convolutions: Although it can be counted as a strength that LBEN can be applied to convnets without much modification, the fact that its performance considerably trails that of MON raises questions about whether the methods presented here are ready to be extended to non-fully connected architectures.
4.	Lack of description of how the Lipschitz bounds of the networks are computed: This critique is self-explanatory.



Decision: I think this paper is well worthy of acceptance just based on the quality and richness of its theoretical development and analysis of LBEN. I’d encourage the authors to, if possible, strengthen the experimental results in directions including (but certainly not limited to) the ones listed above.


Other questions to authors:
1.	I was wondering why you didn’t include experiments involving larger neural networks. What are the limitations (if any) that kept you from trying out larger networks?
2.	Could you describe how you computed the Lipschitz constant? Given how notoriously difficult it is to compute bounds on the Lipschitz constants of neural networks, I think this section requires more elaboration.

Possible typos and minor glitches in writing:
1.	Section 3.2, fist paragraph, first sentence: Should the phrase “equilibrium network” be plural?
2.	D^{+} used in Condition 1 is used before it’s defined in Condition 2.
3.	Just below equation (7): I think there’s a typo in “On the other size, […]”.
4.	In Section 4.1, \epsilon is not used in equation (10), but in equation (11). It might be more clear to introduce \epsilon when (11) is discussed.
5.	Section 4.2, in paragraph “Computing an equilibrium”, first sentence: Do you think there’s a grammar error in this sentence? I might also have mis-parsed the sentence.
6.	Section 5, second sentence: There are two “the”s in a row.




[1] Anil, Cem, James Lucas, and Roger Grosse. "Sorting out lipschitz function approximation." International Conference on Machine Learning. 2019.

---

> ### Author Response · Authors · 2020-11-24
> **Thank you for your comments, in the revised paper we are focusing on improving empirical results.**
>
> We would like to thank the reviewer for their careful reading, positive comments, and helpful feedback. We provide below detailed responses to the list of weak points and questions.
>
> Weak points (using reviewer's numbering):
> 1.  We agree that the submitted paper had quite rudimentary empirical results. Since receiving the reviews, we have been able to generate results with larger convolutional networks with 81 channels on the CIFAR10 dataset. While these are arguably still not ``large-scale'', we have found that in fact we can still achieve both (slightly) improved nominal performance and (dramatically) improved robustness over MON, unconstrained, and feedforward networks. There is no reason in principle why we cannot use our parameterisation with multi-layer convolutional structures and larger models, however it was not feasible for us to explore this fully during the rebuttal period.
> 2. Thank you for this suggestion. We are working on including some kind of comparison to [1], however it can not be a direct comparison since their aim is slightly different: to constrain the Lipschitz constant to be equal to 1, rather enforce upper bounds as a regulariser. Also, in their empirical results they considered Lipschitz continuity with respect to the L-infinity norm, whereas we use L2 norm.
> 3. As mentioned above, since receiving the reviews we have revised the code for convnets, and our new representation now outperforms MON both in terms of nominal performance and robustness to adversarial attack. Full details will appear in the revised paper.
> 4. Thank you for pointing this out, the revised version will have a clearer derivation of the Lipschitz bounds.
>
> Questions:
> 1. Thank you for this question. The computational resource we have for this project at present is a single desktop PC with a single GPU. For the revised version, we have augmented this with a Google Colab instance. However, it still takes several days to generate the results in this paper so expanding to very large networks was not feasible during the rebuttal period.
> 2. In fact, the Lipschitz constant does not need to be computed per se, it is automatically satisfied by our parameterization. The main idea is to consider a dynamical system that has the solution of the LBEN as its equlibrium,  and then compute an incremental $\ell^2$ gain bound of this system using ideas from contraction analysis and integral quadratic constraints. We are working on improving the presentation of this in the revised paper.

---

### Author Response · Authors · 2020-11-25
**Description of main changes in the revised version**

Thanks to all the reviewers for their very helpful comments.

We have submitted a revised manuscript, and here we briefly summarise the main changes:
1. All reviewers requested more empirical results. We have generated results with larger convolutional equilibrium networks on the CIFAR10 data set. We see that the proposed method is superior in terms of both nominal performance and robustness compared to both previous equilibrium network structures and standard feedforward networks of similar sizes.
2. We have added results showing how the quality of the Lipschitz bound varies as a function of network size, examining fully connected networks with widths between 20 and 1000. We see that increasing width leads to only a mild decrease in the quality of the Lipschitz bound, which is encouraging in terms of scalability to very large networks.
3. We have added new material showing how our results can be applied in the special case of feedforward networks, and also shown that our model structure contains all feedforward networks, whereas previously-proposed montone operator equilibrium networks do not.
4. As requested by a reviewer, we have added an illustration of the unperturbed and perturbed images generated by the adversarial attack.
5. We have improved the discussion on connections to convex optimization. We have shown that in the case of ReLU activation, computing the solution of an LBEN corresponds to solving a particular convex quadratic program. We have also fixed an error in the previous version: Prop 4 in the new version requires S=0, this was not stated in the previous version, however it does not effect the main results.
6. We have added a brief discussion to the conclusion about the possibility of extending to different equilibrium network structures.

---

### Decision · Program_Chairs · 2021-01-07
**Final Decision**

**Decision:**

Reject

**Comment:**

This paper is an extension of Monotone Operator Equilibrium Networks (MON). It first tries to address a key issue in MON: whether the activation function $\sigma$ can be represented by a proximal operator of some function $f$. Then it derives the constraints on the weight $W$. Connections to neural ODEs and convex optimization are also built.

Pros:
1. Very nice theory, reads exciting, and provides great insights.
2. Experiments seem to validate part of the theoretical analysis.

Cons: Besides the issue of weak experiments raised by the reviewers (and the authors admitted "quite rudimentary empirical results"), the AC regretted that the proofs have some key flaws.
1. The proof of Proposition 1, which is the cornerstone of the paper, is wrong. Although the AC believed that Proposition 1 is likely to be true, the current proof of "only if" is unfortunately incorrect. Please notice the claim at the first line of the "only if" proof, which writes "$\sigma$ is a non-decreasing and piece-wise differentiable function". Although it is known in real analysis that monotone functions are almost everywhere differentiable, this does not necessarily mean that monotone functions can be piece-wise differentiable functions as the non-differentiable points can be dense, though of zero measure.
2. Proposition 7, which claims that LBEN parameterization (8) contains all feedforward networks, is wrong. From the proof on page 19, the AC doubted whether the lower diagonal blocks of $-2H_D^{-1}  H_L$ can be $\{W_i\}$. This is because by the definition of $H$ given at 5 lines below (40), $H$ must be positive definite. If the lower diagonal blocks of $-2H_D^{-1} H_L$ can be $\{W_i\}$, then $H_i=\Lambda_i W_i$.  It is not apparent whether for any choice of $\{W_i\}$ there exist $\{\Lambda_i\} \in D^+$, such that $H$ is positive definite. Moreover, the lower diagonal block of $V^TV$, i.e., $\Gamma_{j+1}V_j\Phi_j$ (corrected from the typo in the minor issue 2 below), must equal to $\Lambda_j W_j$. However, it is obvious that for some $W_j$ (e.g., it has more columns than rows), there cannot exist $\Gamma_{j+1}$, $V_j$, $\Phi_j$ and $\Lambda_j$, such that $\Gamma_{j+1}V_j\Phi_j=\Lambda_j W_j$, due to the structures of these matrices. Moreover, due to the existence of $(...)/(2\gamma)$ in the definition of $H$, the diagonal blocks of $H$ cannot be diagonal by the choice of $V$.
3. Proposition 7 again. The choice of parameters in (44) does not match the feedforward network in (15). Please observe that by (44) $z_1=\sigma(U_0 x + b_0)$, rather than $z_1=U_0 x + b_0$ in (15).

Minor issues:
1. The proof on page 19 is for claiming that (9), rather than (8), contains all feedforward networks.
2. In the choice of $V$, $\Psi_i V_i$ should be $V_i\Phi_i$.

Although the reviewers and the AC liked the paper, due to the above flaws and limited acceptance rate, the AC deemed that the flaws should be fixed prior to acceptance.